# Covariance-Robust Minimax Probability Machines for Algorithmic Recourse

## Abstract

Algorithmic recourse is rising as a prominent technique to promote the explainability and transparency of the predictive model in ethical machine learning. Existing approaches to algorithmic recourse often assume an invariant predictive model; however, this model, in reality, is usually updated temporally upon the input of new data. Thus, a recourse that is valid respective to the present model may become *in*valid for the future model. To resolve this issue, we propose a pipeline to generate a model-agnostic recourse that is robust to model shifts. Our pipeline first estimates a linear surrogate of the *non*linear (black-box) model using covariance-robust minimax probability machines (MPM); then, the recourse is generated with respect to this robust linear surrogate. We show that the covariance-robust MPM recovers popular regularization schemes, including $\ell_2$-regularization and class-reweighting. We also show that our covariance-robust MPM pushes the decision boundary in an intuitive manner, which facilitates an interpretable generation of a robust recourse. The numerical results demonstrate the usefulness and robustness of our pipeline.

## 1 Introduction

The recent prevalence of machine learning (ML) in supporting consequential decisions involving humans such as loan approval (Moscato et al., 2021), job hiring (Cohen et al., 2019; Schumann et al., 2020), and criminal justice (Brayne & Christin, 2021) urges the need of transparent ML systems with explanations and feedback to users (Doshi-Velez & Kim, 2017; Miller, 2019). One popular and emerging approach to providing feedback is the algorithmic recourse (Ustun et al., 2019). A *recourse* suggests how the input instance should be modified to alter the outcome of a predictive model. Consider a specific scenario in which an individual is rejected from receiving a loan by a financial institution's ML model. Recently, it has become a legal necessity to provide explanations and recommendations to the individual so that they can improve their situation and obtain a loan in the future (GDPR, Voigt & Von dem Bussche (2017)). For example, an explanation can be "increase the income to \$5000" or "reduce the debt/asset ratio to below 20%". Leveraging the recourses, financial institutions can assess the reliability of their ML predictive models and increase user engagement through actionable feedback and acceptance guarantee if they fulfill the requirements.

To construct plausible and meaningful recourses, one must assess and strike a balance between conflicting criteria. They can be: (1) *validity*, a recourse should effectively reverse the unfavorable prediction of the model into a favorable one, (2) *proximity*, recourse should be close to the original input instance to alleviate the efforts required, and thus to encourage the adoption of the recourse, (3) *actionability*, prescribed modifications should follow causal laws of our society (Ustun et al., 2019; Karimi et al., 2021); for example, one can not modify their race or decrease their age.

Various techniques were proposed to devise algorithmic recourses for a given predictive model, extensive surveys are provided in (Karimi et al., 2020a; Stepin et al., 2021; Pawelczyk et al., 2021; Verma et al., 2020). Wachter et al. (2017) introduced the definition of counterfactual explanations and proposed a gradient-based approach to find the nearest instance that yields a favorable outcome. Ustun et al. (2019) proposed a mixed integer programming formulation (AR) that can find recourses for a linear classifier with a flexible design of the actionability constraints. Alternatively, Karimi et al. (2021; 2020b) investigated the nearest recourse through the lens of minimal intervention to take causal relationships between features into account. Recent works including Russell (2019)

and Mothilal et al. (2020) also studied the problem of generating a menu of diverse recourses to provide multiple possibilities that users might choose.

The aforementioned methods rely on an assumption of an invariant predictive model. Nevertheless, machine learning models are usually re-trained or re-calibrated as new data arrive. Thus, a valid recourse at present may become invalid in the future, leading to an exemplary case where a rejected applicant may spend efforts to improve their income and reapply for a loan, but then is rejected (again) simply because the ML model has been updated. This leads to a potential inefficiency due to the waste of resources and loss of trust in the recommendation and in the ML system (Rudin, 2019).

Studying this phenomenon, Rawal et al. (2020) described several types of model shifts related to the correction, temporal, and geospatial shifts from data. They pointed out that the recourses, even constructed with state-of-the-art algorithms, are vulnerable to distributional shifts in the model's parameters. Pawelczyk et al. (2020) study counterfactual explanations under predictive multiplicity and its relation to the difference in the way two classifiers treat predicted individuals. Black et al. (2021) then show that the constructed recourses might be invalid even for the model retrained with different initial conditions such as weight initialization and leave-one-out variations in data. Recently, Upadhyay et al. (2021) leveraged robust optimization to propose ROAR - a framework for generating recourses that are robust to shifts in the predictive model, which is assumed to be a linear classifier.

Despite the promising results, existing methods are often restricted to the linear classifiers setting to be able to introduce actionability or robustness (Ustun et al., 2019; Russell, 2019; Upadhyay et al., 2021; Rawal et al., 2020). For non-linear classifiers, a linear surrogate method such as LIME (Ribeiro et al., 2016) is used to approximate the local decision boundary of the black-box classifiers; the recourse is then generated respectively to the (linear) surrogate model instead of the nonlinear model. LIME is well-known for explaining predictions of black-box ML models by fitting a reweighted linear regression model to the perturbed samples around an input instance. In the recourse literature, LIME is the most common linear surrogate for the local decision boundary of the black-box models (Ustun et al., 2019; Upadhyay et al., 2021).

Unfortunately, the LIME surrogate has several limitations. Firstly, Laugel et al. (2018) and White & Garcez (2019) showed that LIME may not be faithful to the underlying models because LIME might be influenced by input features at a global scale rather than a local scale. Secondly, explanations generated by perturbation-based methods are also well-known to be sensitive to the original input and the synthesized perturbations (Alvarez-Melis & Jaakkola, 2018; Ghorbani et al., 2019; Slack et al., 2020; 2021; Agarwal et al., 2021; Laugel et al., 2018).

Several works have been proposed to overcome these issues. Laugel et al. (2018) and Vlassopoulos et al. (2020) proposed alternative sampling procedures that generate sample instances in the neighborhood of the closest counterfactual to fit a local surrogate. White & Garcez (2019) integrated counterfactual explanation to local surrogate models to introduce a novel fidelity measure of an explanation. Later, Garreau & von Luxburg (2020) and Agarwal et al. (2021) analyzed theoretically the stability[1] of LIME, especially in the low sampling size regime. Zhao et al. (2021) leveraged Bayesian reasoning to improve the consistency in repeated explanations of a single prediction. Nevertheless, the impact and effectiveness of these surrogates on the recourse generation are still unknown.

**Contributions.** We revisit the recourse generation scheme through surrogate models. We propose a novel model-agnostic pipeline that facilitates the generation of *robust* and *actionable* recourses. The core innovation in our pipeline is the use of the covariance-robust minimax probability machines (MPM) as a linear surrogate of the nonlinear black-box ML model. Additionally, we contribute

- to the field of MPM and robust classifier: We propose and analyze in detail the covariance-robust MPMs in which the set of possible perturbations of the covariance matrices are prescribed using distances on the space of positive semidefinite matrices. Motivated by the statistical distances between Gaussian distributions, we show that the covariance-robustness induces and connects to two prominent regularization schemes of the nominal MPM:
  - if the distance is motivated by the Bures distance, we recover the $\ell_2$-regularization,
  - if the distance is motivated by the Fisher-Rao distance, we recover class reweighting schemes.

---

[1] Throughout, "robustness" is used in the algorithmic recourse setting with respect to the model shifts (Rawal et al., 2020). "Robustness" is also used to indicate the sensitivity of LIME to the sampling distribution. To avoid confusion, in what follows, we use "stability" to refer to the aforementioned sensitivity of LIME.

Figure 1: The sampler synthesizes new instances around $x_0$ and queries the predicted labels from the classifier $f$ . The moment information $(\widehat{\mu}_y, \widehat{\Sigma}_y)$ estimated from the synthetic psedo-labeled data (represented by triangles and ellipsoids) serves as inputs for the Covariance-robust MPM. The MPM surrogate $\theta^\varphi$ (red hyperplane) is the target classifier used to generate recourses (red circle).

While prior works showed that distributionally robust optimization (DRO) with optimal transport distance recovers norm regularization (Shafieezadeh-Abadeh et al., 2019b) and $f$-divergence DRO leads to reweighting (Duchi & Namkoong, 2019), this paper extends the connections to the MPMs.

- to the field of robust algorithmic recourse: We propose an intuitive and interpretable approach to generate robust recourse. We show that, by calibrating the radii of the ambiguity sets in a proper manner, the covariance-robust MPM shifts the separation hyperplane towards the favorable class. As a consequence, our recourse exhibits robustness to model shifts and it is also lenient to incorporate mixed-integer constraints to promote actionability.

This paper unfolds as follows. In Section 2, we delineate our explanation framework using MPM. Section 3 dives deeper into the MPM problem and its robustification. Section 4-5 construct two types of covariance-robust MPM using the Bures and Fisher-Rao distance on the space of covariance matrices. In Section 6, we demonstrate empirically that the covariance-robust MPM provides a competitive approximation of the local decision boundary, and improves the robustness of the recourse subject to model shifts. All proofs are relegated to the appendix.

## 2 RECOURSE GENERATION FRAMEWORK

Throughout this paper, we assume that the covariate space is $\mathcal{X} = \mathbb{R}^d$ and we have a binary label space $\mathcal{Y} = \{-1, +1\}$. Without any loss of generality, we assume that label $-1$ encodes the unfavorable decision, while $+1$ encodes the favorable one. Given a specific classifier and an input $x_0$ with an unfavorable predicted outcome, the goal of this paper is to find a recourse recommendation for $x_0$ that has a high probability of being classified into a favorable group, subject to possible shifts in the parameters underlying the classifier. Such recourse is termed a robust recourse. Our robust recourse generator consists of three components (see Figure 1 for a schematic view):

(i) a local sampler: we use a similar procedure as in Vlassopoulos et al. (2020) and Laugel et al. (2018). Given an instance $x_0$, we choose $k$ nearest counterfactuals $x_1, \ldots, x_k$ from the training data that have the opposite label to $x_0$. For each $x_i$, we perform a line search to find a point $x_{b,i}$ that is on the decision boundary and the line segment between $x_0$ and $x_i$. Among $x_{b,i}$, we choose the nearest point $x_b$ to $x_0$ and sample uniformly in an $\ell_2$-neighborhood with radius $r_p$ around $x_b$. We then query the black-box classifier to obtain the predicted labels of the synthetic samples.

(ii) a linear surrogate using (covariance-robust) MPM: We use the synthetic samples to estimate the moment information $(\widehat{\mu}_y, \widehat{\Sigma}_y)$ of the covariate conditional on each predicted class $y$. We then train a covariance-robust MPM parametrized by $\theta^\varphi$ to approximate the local decision boundary of the ML model.

(iii) a recourse search: Basically, we can apply any existing recourse search for linear models on top of the linear surrogate $\theta^\varphi$ dictated by the covariance-robust MPMs to find a *robust* recourse. In this paper, we use a simple projection onto the hyperplane prescribed by $\theta^\varphi$ for simplicity and AR (Ustun et al., 2019), which is a MIP-based framework, to promote *actionable* recourses.

Center to the success of our pipeline is the possibility of *shifting* the MPM classification hyperplane toward the region of the favorable class, which induces robust recourse with respect to model shifts in a geometrically-intuitive manner (see Remark 4.6 for a detailed discussion). It is imperative to

note a clear distinction between our pipeline and the existing method of ROAR (Upadhyay et al., 2021): ROAR uses a non-robust surrogate in Step (ii) and then formulates a min-max optimization problem in Step (iii) for recourse search, whilst our pipeline uses a *robust* surrogate in Step (ii) and then employs a simple recourse search in Step (iii). Note that mixed-integer formulations can be injected in Step (iii) to generate more realistic robust recourses in our pipeline. On the contrary, mixed-integer constraints are not easy to be integrated into the min-max formulation of ROAR.

Subsequently, Sections 3-5 describe in detail different methods to build robust surrogates in Step (ii). The application of recourse research in Step (iii) is provided in the experiment section (Section 6.2).

## 3 (COVARIANCE-ROBUST) MPM

MPM is a binary classification framework pioneered by Lanckriet et al. (2001) and extended to Quadratic MPM in (Lanckriet et al., 2003). For each class $y \in \mathcal{Y}$, MPM makes no assumption on the specific parametric form of the (conditional) distribution $\widehat{\mathbb{P}}_y$ of $X|Y = y$. Instead, MPM assumes that we can identify $\widehat{\mathbb{P}}_y$ only up to the first two moments, *i.e.*, it assumes that $\widehat{\mathbb{P}}_y$ has mean vector $\widehat{\mu}_y \in \mathbb{R}^d$ and covariance matrix $\widehat{\Sigma}_y \in \mathbb{S}_+^d$, denoted $\widehat{\mathbb{P}}_y \sim (\widehat{\mu}_y, \widehat{\Sigma}_y)$. These moments can be estimated from the samples synthesized from the boundary sampler. The goal of MPM is to find a (non-trivial) linear classifier that minimizes the maximum misclassification rate among classes. To this end, we consider the family of linear classifiers parametrized by $\theta = (w, b) \in \mathbb{R}^{d+1}$, $w \neq 0$ with classification rule

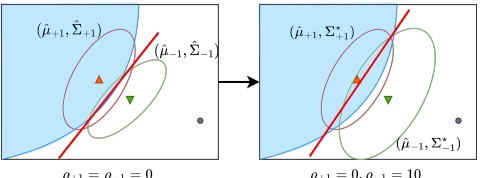

Figure 2: An intuitive explanation of the robustification mechanism. From left to right: As the radius $\rho_{-1}$ increases, the worst-case covariance matrix of the class $-1$ is inflated and shifts the MPM boundary towards the favorable class. The projection of the input $x_0$ onto the hyperplane will have a tendency to lie deeper into the favorable region and may become more robust to model shifts.

$$\mathcal{C}_\theta(x) = \mathrm{sign}(w^\top x - b),$$

where $(w, b)$ is the slope and intercept. The MPM solves the min-max optimization problem

$$\min_{\theta \in \Theta} \max_{y \in \mathcal{Y}} \max_{\widehat{\mathbb{P}}_y \sim (\widehat{\mu}_y, \widehat{\Sigma}_y)} \widehat{\mathbb{P}}_y(\mathcal{C}_\theta(X) \neq y), \tag{1}$$

where we define the feasible set $\Theta \triangleq \{\theta = (w, b) \in \mathbb{R}^{d+1} : w \neq 0\}$. Notice that the constraints $w \neq 0$ eliminate trivial solutions to the classification problem.

To derive the MPM, we define the set of feasible slopes $\mathcal{W} = \left\{ w \in \mathbb{R}^d \backslash \{0\} : \sum_{y \in \mathcal{Y}} y w^\top \widehat{\mu}_y = 1 \right\}$, which is a hyperplane in $\mathbb{R}^d$. The main instrument for solving (1) is the following result from (Lanckriet et al., 2001, §2) which provides the form of its optimal solution.

**Lemma 3.1** (Optimal solution). *Let $\widehat{w}$ be an optimal solution to the second-order cone program*

$$\min_{w \in \mathcal{W}} \sum_{y \in \mathcal{Y}} \sqrt{w^\top \widehat{\Sigma}_y w}, \tag{2}$$

*then $\widehat{\theta} = (\widehat{w}, \widehat{b})$ solves the MPM problem* (1), *where $\widehat{\kappa} = \left( \sum_{y \in \mathcal{Y}} \sqrt{\widehat{w}^\top \widehat{\Sigma}_y \widehat{w}} \right)^{-1}$, and $\widehat{b} = \widehat{w}^\top \widehat{\mu}_{+1} - \widehat{\kappa}\sqrt{\widehat{w}^\top \widehat{\Sigma}_{+1} \widehat{w}}$.*

In this paper, we refer to the second-order cone program (2) as the *nominal* MPM problem, because the MPM is fully determined by the solution to (2). We next discuss the covariance-robust MPM.

### 3.1 QUADRATIC MPM

In a practical setting, it is likely that the covariance matrices $\widehat{\Sigma}_y$ are misspecified, for example, due to low sample size, statistical estimation error, or corrupted data. To hedge against these mismatches, Lanckriet et al. (2003) proposed to add another layer of robustness by allowing the mean vectors and the covariance matrices of the conditional distributions to be chosen (adversarially) in a prescribed set,

which we call the ambiguity set. They showed that perturbing the mean vectors does not change the optimal classifier. In this paper, we, therefore, perturb only the covariance matrices. More specifically, we allow the conditional distribution $\mathbb{P}_y$ to be in the ambiguity set

$$\mathbb{U}_y^\varphi(\widehat{\mathbb{P}}_y) = \{\mathbb{P}_y : \mathbb{P}_y \sim (\widehat{\mu}_y, \Sigma_y), \ \varphi(\Sigma_y \parallel \widehat{\Sigma}_y) \le \rho_y\},$$

where $\varphi$ is a measure of dissimilarity between covariance matrices. The distributionally robust minimax probability machine is formulated as

$$\min_{\theta \in \Theta} \ \max_{y \in \mathcal{Y}} \ \max_{\mathbb{P}_y \in \mathbb{U}_y^\varphi(\widehat{\mathbb{P}}_y)} \ \mathbb{P}_y(\mathcal{C}_\theta(X) \ne y). \tag{3}$$

Previously, Lanckriet et al. (2003) considered the robust MPM with moment uncertainty, in which the covariance matrix is perturbed using the quadratic divergence.

**Definition 3.2** (Quadratic divergence). *Given two positive semidefinite matrices $\Sigma$, $\widehat{\Sigma} \in \mathbb{S}_+^d$, the quadratic divergence between them is $\mathbb{Q}(\Sigma \parallel \widehat{\Sigma}) = \mathrm{Tr}\left[(\Sigma - \widehat{\Sigma})^2\right]$.*

The divergence $\mathbb{Q}$ is the *squared* Frobenius norm of $\Sigma - \widehat{\Sigma}$; thus $\mathbb{Q}$ is non-negative and vanishes to zero if and only if $\Sigma = \widehat{\Sigma}$, so it is a divergence on $\mathbb{S}_+^d$. The Quadratic MPM has the below form (Lanckriet et al., 2003).

**Theorem 3.3** (Quadratic MPM). *Suppose that $\varphi \equiv \mathbb{Q}$. Let $w^{\mathbb{Q}}$ be a solution to the problem*

$$\min_{w \in \mathcal{W}} \ \sum_{y \in \mathcal{Y}} \sqrt{w^\top (\widehat{\Sigma}_y + \sqrt{\rho_y} I) w}. \tag{4}$$

*Then $\theta^{\mathbb{Q}} = (w^{\mathbb{Q}}, b^{\mathbb{Q}})$ solves the distributionally robust MPM problem* (3)*, with*

$$\kappa^{\mathbb{Q}} = \left( \sum_{y \in \mathcal{Y}} \sqrt{(w^{\mathbb{Q}})^\top (\widehat{\Sigma}_y + \sqrt{\rho_y} I) w^{\mathbb{Q}}} \right)^{-1}, \quad b^{\mathbb{Q}} = (w^{\mathbb{Q}})^\top \widehat{\mu}_{+1} - \kappa^{\mathbb{Q}} \sqrt{(w^{\mathbb{Q}})^\top (\widehat{\Sigma}_{+1} + \sqrt{\rho_{+1}} I) w^{\mathbb{Q}}}.$$

The Quadratic MPM can be considered as a regularization of the nominal problem (2): each matrix $\widehat{\Sigma}_y$ is added with a diagonal matrix $\sqrt{\rho_y} I$, making the matrix better conditioned. This is equivalently known as inverse regularization, which ensures invertibility when $\widehat{\Sigma}_y$ is low-rank and $\rho_y > 0$.

## 3.2 COVARIANCE-ROBUST MPM

While Lanckriet et al. (2003) focused only on the quadratic divergence, their results can be generalized to the covariance-robust MPM with a general divergence $\varphi$. For any $y \in \mathcal{Y}$, define $\tau_y : \mathbb{R}^d \to \mathbb{R}$ as

$$\tau_y^\varphi(w) \triangleq \max_{\Sigma_y \in \mathbb{S}_+^d : \varphi(\Sigma_y \parallel \widehat{\Sigma}_y) \le \rho_y} \sqrt{w^\top \Sigma_y w}. \tag{5}$$

We are now ready to give a generalized reformulation of problem (3).

**Proposition 3.4** (Covariance-robust MPM). *Let $w^\varphi$ be the optimal solution to the problem*

$$\min_{w \in \mathcal{W}} \ \sum_{y \in \mathcal{Y}} \tau_y^\varphi(w),$$

*then $\theta^\varphi = (w^\varphi, b^\varphi)$ solves the distributionally robust MPM problem* (3)*, where*

$$\kappa^\varphi = \left( \sum_{y \in \mathcal{Y}} \tau_y^\varphi(w^\varphi) \right)^{-1}, \quad and \quad b^\varphi = (w^\varphi)^\top \widehat{\mu}_{+1} - \kappa^\varphi \tau_{+1}^\varphi(w^\varphi).$$

## 3.3 EQUIVALENCE UNDER GAUSSIAN ASSUMPTIONS

While the quadratic divergence $\mathbb{Q}$ in Definition 3.2 is attractive for its tractability, it is not statistically meaningful. More specifically, it does not coincide with any distance between probability distributions with the corresponding covariance information. In this paper, we consider discrepancy measures $\varphi$ that arise as a statistical distance between Gaussian distributions. To this goal, we first need to show that the covariance-robust MPM is invariant with the Gaussian assumption. Define a parametric ambiguity set constructed on the space of Gaussian distribution of the form

$$\mathcal{U}_y^\mathcal{N}(\widehat{\mathbb{P}}_y) = \left\{ \mathbb{P}_y \in \mathcal{P}(\mathcal{X}) : \ \mathbb{P}_y \sim \mathcal{N}(\widehat{\mu}_y, \Sigma_y), \varphi(\Sigma_y \parallel \widehat{\Sigma}_y) \le \rho_y \right\},$$

wherein any distribution is Gaussian. Consider the Gaussian distributionally robust MPM problem

$$\min_{\theta} \ \max_{y \in \mathcal{Y}} \ \max_{\mathbb{P}_y \in \mathcal{U}_y^\mathcal{N}(\widehat{\mathbb{P}}_y)} \ \mathbb{P}_y(\mathcal{C}_\theta(X) \ne y). \tag{6}$$

**Proposition 3.5** (Gaussian equivalence). *The optimizer $\theta^\varphi = (w^\varphi, b^\varphi)$ in Proposition 3.4 also solves the Gaussian parametric covariance-robust MPM problem* (6).

Proposition 3.5 justifies the use of divergences induced by a distance between normal distributions. We study several constructions of the covariance-robust MPM in the subsequent sections.

## 4 BURES MPM

We first explore the case where $\varphi$ is the Bures divergence whose definition is as follows.

**Definition 4.1** (Bures divergence). *Given two positive semi-definite matrices $\Sigma, \widehat{\Sigma} \in \mathbb{S}_+^d$, the Bures divergence between them is $\mathbb{B}(\Sigma \parallel \widehat{\Sigma}) = \mathrm{Tr}\left[\Sigma + \widehat{\Sigma} - 2(\widehat{\Sigma}^{\frac{1}{2}}\Sigma\widehat{\Sigma}^{\frac{1}{2}})^{\frac{1}{2}}\right]$.*

It can be shown that $\mathbb{B}$ is symmetric and non-negative, and it vanishes to zero if and only if $\Sigma = \widehat{\Sigma}$. As such, $\mathbb{B}$ is a divergence on the space of positive semidefinite matrices. Moreover, $\mathbb{B}$ also equals the *squared* type-2 Wasserstein distance between two Gaussian distributions with the same mean vector and covariance matrices $\Sigma$ and $\widehat{\Sigma}$ (Olkin & Pukelsheim, 1982; Givens & Shortt, 1984; Gelbrich, 1990). Next, we assert the form of the Bures MPM.

**Theorem 4.2** (Bures MPM). *Suppose that $\varphi \equiv \mathbb{B}$. Let $w^\mathbb{B}$ be the solution of the following problem*

$$\min_{w \in \mathcal{W}} \sum_{y \in \mathcal{Y}} \sqrt{w^\top \widehat{\Sigma}_y w} + \big(\sum_{y \in \mathcal{Y}} \rho_y\big)\|w\|_2. \tag{7}$$

*Then $\theta^\mathbb{B} = (w^\mathbb{B}, b^\mathbb{B})$ is the optimal solution of the distributionally robust MPM problem* (3)*, where*

$$\kappa^\mathbb{B} = \left(\sum_{y \in \mathcal{Y}} \sqrt{(w^\mathbb{B})^\top \widehat{\Sigma}_y w^\mathbb{B}} + \big(\sum_{y \in \mathcal{Y}} \rho_y\big)\|w^\mathbb{B}\|_2\right)^{-1}, \quad and$$

$$b^\mathbb{B} = (w^\mathbb{B})^\top \widehat{\mu}_{+1} - \kappa^\mathbb{B}\left(\sqrt{(w^\mathbb{B})^\top \widehat{\Sigma}_{+1} w^\mathbb{B}} + \rho_{+1}\|w^\mathbb{B}\|_2\right).$$

Theorem 4.2 unveils a fundamental connection between robustness and regularization: if we construct the ambiguity sets for the covariance matrices using the Bures divergence, the resulting optimization problem (7) is an $l_2$-regularization of the nominal problem (2). This connection aligns with previous observations highlighting the equivalence between regularization schemes and optimal transport robustness (Shafieezadeh-Abadeh et al., 2019a; Blanchet et al., 2019).

To prove Theorem 4.2, we provide a result that asserts the analytical form of $\tau_y^\mathbb{B}(w)$.

**Proposition 4.3** (Bures divergence). *If $\varphi \equiv \mathbb{B}$, then $\tau_y^\mathbb{B}(w) = \rho_y\|w\|_2 + \sqrt{w^\top \widehat{\Sigma}_y w}$ for all $y \in \mathcal{Y}$.*

The proof of Theorem 4.2 follows by combining Propositions 3.4 and 4.3.

Next, we study the asymptotic form of the Bures MPM as the radii of the ambiguity sets grow. Note that problem (7) depends only on the *sum* of the radii, but not on the individual values of each radius. Let $\rho = \sum_{y \in \mathcal{Y}} \rho_y$ be their sum, it suffices to study when $\rho$ grows to infinity. To this end, denoted by $w_\rho^\mathbb{B}$ the optimal solution of problem (7) parametrized by $\rho$. The next result provides the analysis of the asymptotic value of $w_\rho^\mathbb{B}$ as $\rho \to \infty$.

**Proposition 4.4** (Bures asymptotic hyperplane). *Fix $y \in \mathcal{Y}$, let $-y$ be its opposite class and suppose that $\rho_{-y}$ remains constant. Let $w_{\rho_y}^\mathbb{B}$ be the optimal solution of (7) parametrized by $\rho_y$. As $\rho_y \to \infty$,*

$$w_{\rho_y}^\mathbb{B} \to w_\infty^\mathbb{B} \triangleq \big(\sum_{y \in \mathcal{Y}} y\widehat{\mu}_y\big)\big/\|\sum_{y \in \mathcal{Y}} y\widehat{\mu}_y\|_2^2.$$

Notice that as $\rho_y \to \infty$, $\kappa^\mathbb{B}\tau_y^\mathbb{B}(w_\infty^\mathbb{B}) \to 1$. Thus, $b_\infty^\mathbb{B} = (w_\infty^\mathbb{B})^\top \widehat{\mu}_y - y$ for any $y \in \mathcal{Y}$. The asymptotic hyperplane defined by $w_\infty^\mathbb{B}$ is thus characterized by the linear equation $w_\infty^\mathbb{B} x - w_\infty^\mathbb{B} \widehat{\mu}_y + y = 0$, which identifies a hyperplane passing through $\widehat{\mu}_{-y}$ as $\sum_{y \in \mathcal{Y}} y w^\top \mu_y = 1$. Moreover, we note that the asymptotic hyperplane does not depend on the covariance matrices.

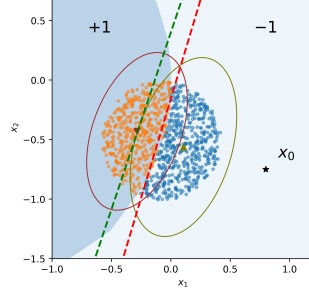

Figure 3: A 2D example of the Bures MPM hyperplanes with fixed $\rho_{+1} = 0$ and $\rho_{-1} = 0.01$ (red) and $\rho_{-1} = 10$ (green). The green line is pushed towards the favorable region (predicted as +1).

**Remark 4.5** (Quadratic asymptotic hyperplane). *It is provable that the solution of problem* (4) *converges to $w_\infty^{\mathrm{B}}$ as $\rho = \sum_y \rho_y$ tends to infinity. Thus, Quadratic MPM* (4) *and Bures MPM* (7) *are asymptotically equivalent even though they induce different regularizations of the nominal MPM* (2).

**Remark 4.6** (Geometric intuition for robust recourse). *Figure 3 visualizes the Bures MPM hyperplanes by varying the radii. Notice that the hyperplane drifts toward the favorable (+1) class as the uncertainty of the unfavorable (-1) covariance matrix increases. Thus, the recourse generated w.r.t. the green hyperplane is more robust compared to that generated w.r.t. the red one. By calibrating the radii, we shift the hyperplane and obtain robust recourses at different robustness-cost trade-offs.*

## 5 FISHER-RAO MPM

We now explore the case where $\varphi$ is the Fisher-Rao distance which is defined as follows.

**Definition 5.1** (Fisher-Rao distance). *Given two positive definite matrices $\Sigma, \widehat{\Sigma} \in \mathbb{S}_{++}^d$, the Fisher-Rao distance between them is $\mathbb{F}(\Sigma, \widehat{\Sigma}) = \| \log(\widehat{\Sigma}^{-\frac{1}{2}} \Sigma \widehat{\Sigma}^{-\frac{1}{2}}) \|_2$, where $\log(\cdot)$ is the matrix logarithm.*

The Fisher-Rao distance enjoys many nice properties. In particular, it is invariant to inversion and congruence, *i.e.*, for any $\Sigma, \widehat{\Sigma} \in \mathbb{S}_{++}^d$ and invertible $A \in \mathbb{R}^{d \times d}$, $\mathbb{F}(\Sigma, \widehat{\Sigma}) = \mathbb{F}(\Sigma^{-1}, \widehat{\Sigma}^{-1}) = \mathbb{F}(A\Sigma A^\top, A\widehat{\Sigma} A^\top)$. Such invariances are especially statistically meaningful as it implies that the results remain unchanged if we reparametrize the problem with an inverse covariance matrix (instead of the covariance matrix) or apply a change of basis to the data space $\mathcal{X}$. It is shown that $\mathbb{F}$ is the unique Riemannian distance (up to scaling) on the cone $\mathbb{S}_{++}^d$ with such invariances (Savage, 1982). Next, we assert the form of the Fisher-Rao MPM.

**Theorem 5.2** (Fisher-Rao MPM). *Suppose that $\varphi \equiv \mathbb{F}$. Let $w^{\mathbb{F}}$ be the solution of the problem*

$$\min_{w \in \mathcal{W}} \sum_{y \in \mathcal{Y}} \exp\left(\frac{\rho_y}{2}\right) \sqrt{w^\top \widehat{\Sigma}_y w}. \tag{8}$$

*Then $\theta^{\mathbb{F}} = (w^{\mathbb{F}}, b^{\mathbb{F}})$ is the optimal solution of the distributionally robust MPM problem* (3), *where*

$$\kappa^{\mathbb{F}} = \left( \sum_{y \in \mathcal{Y}} \exp\left(\frac{\rho_y}{2}\right) \sqrt{(w^{\mathbb{F}})^\top \widehat{\Sigma}_y w^{\mathbb{F}}} \right)^{-1}, \text{ and } b^{\mathbb{F}} = (w^{\mathbb{F}})^\top \widehat{\mu}_{+1} - \kappa^{\mathbb{F}} \exp\left(\frac{\rho_{+1}}{2}\right) \sqrt{(w^{\mathbb{F}})^\top \widehat{\Sigma}_{+1} w^{\mathbb{F}}}.$$

Theorem 5.2 divulges another foundational connection between robustness and regularization: if we construct the ambiguity sets for the covariance matrices using the Fisher-Rao distance, the resulting optimization problem (8) is a *reweighted* version of the nominal problem (2). Each term $(w^\top \widehat{\Sigma}_y w)^{\frac{1}{2}}$ is assigned a weight $\exp(\rho_y/2)$, which is proportional to the radius $\rho_y$. This connection aligns with previous observations highlighting the equivalence between reweighting schemes and distributional robustness (Ben-Tal et al., 2013; Bayraksan & Love, 2015; Namkoong & Duchi, 2017; Hashimoto et al., 2018). To prove Theorem 5.2, we derive an analytical expression of $\tau_y^{\mathbb{F}}(w)$.

**Proposition 5.3** (Fisher-Rao distance). *If $\varphi \equiv \mathbb{F}$, then $\tau_y^{\mathbb{F}}(w) = \exp\left(\frac{\rho_y}{2}\right)(w^\top \widehat{\Sigma} w)^{\frac{1}{2}}$ for all $y \in \mathcal{Y}$.*

The proof of Theorem 5.2 follows by combining the results from Proposition 3.4 and Proposition 5.3. Next, we study the asymptotic form of the Fisher-Rao MPM.

**Proposition 5.4** (Fisher-Rao asymptotic hyperplane). *Fix $y \in \mathcal{Y}$, let $-y$ be its opposite class and suppose that $\rho_{-y}$ remains constant. Let $w_{\rho_y}^{\mathbb{F}}$ be the optimal solution of* (8) *parametrized by $\rho_y$. Let $a \triangleq \sum_{y \in \mathcal{Y}} y\widehat{\mu}_y$, then as $\rho_y \to \infty$, $w_{\rho_y}^{\mathbb{F}} \to w_{\infty,y}^{\mathbb{F}} \triangleq (a^\top \widehat{\Sigma}_y^{-1} a)^{-1} \widehat{\Sigma}_y^{-1} a$.*

Contrary to the Bures MPM, the asymptotic hyperplane of Fisher-Rao MPM depends explicitly on the covariance matrix $\widehat{\Sigma}_y$. The boundary prescribed by the Fisher-Rao MPM can be shifted through an appropriate calibration of the radii $\rho_y$. Thus, Fisher-Rao MPM can generate a robust algorithmic recourse in a geometrically-intuitive manner.

## 6 NUMERICAL EXPERIMENTS

We conduct comprehensive experiments to highlight the performance of our models. We first compare the fidelity and stability of our surrogates with LIME. We then compare the quality of our recourses against two popular baselines: ROAR (Upadhyay et al., 2021) and AR (Ustun et al., 2019).

**Classifier.** We use three-layer MLP with 20, 50, 20 nodes and ReLU activation functions in each consecutive layer. We use a sigmoid function in the last layer to produce probabilities.

**Dataset.** We evaluate our framework using popular real-world datasets for algorithmic recourse: *German Credit* (Dua & Graff, 2017; Groemping, 2019), *Small Bussiness Administration (SBA)* (Li et al., 2018), and *Student performance* (Cortez & Silva, 2008). Each dataset contains two sets of data (the present data - $D_1$ and the shifted data $D_2$). The shifted dataset $D_2$ could capture the correction shift (German credit), the temporal shift (SBA), or the geospatial shift (Student). For each dataset, we use 80% of the instances in the present data $D_1$ to train an underlying predictive model and the remaining instances are used to generate and evaluate recourses. The shifted data $D_2$ is used to train future classifiers in Section 6.2. The main text contains the results for the German and Student datasets. Further results, including the SBA and the synthetic data, are provided in Appendix A.

## 6.1 FIDELITY AND STABILITY OF THE SURROGATE MODELS

We evaluate the performance of different surrogate models with respect to the current classifier. We compare our methods: QUAD-MPM in (3.3), the BW-MPM in (4.2) and the FR-MPM in (5.2) against the popular linear surrogate LIME (Ribeiro et al., 2016) under the following metrics:

**Stability.** We use the procedure in Agarwal et al. (2021) to measure the stability of the surrogate models with respect to small perturbations in the input instance. For a given instance $x$, we draw a set $\mathcal{U}_x$ of 10 neighbors of $x$ from $\mathcal{N}(x, 0.001I)$ independently. We use the above-mentioned methods to find the linear surrogate $\theta_{x'} = (w_{x'}, b_{x'})$ for each $x' \in \mathcal{U}_x$. We report the maximum distance of the explanations of $x'$ to that of $x$; precisely, the stability formula is

$$\text{Stability}(w_x) = \max_{x' \in \mathcal{U}_x} \|w_x - w_{x'}\|_2.$$

**Fidelity.** We use the LocalFid criterion as in Laugel et al. (2018) to measure the fidelity of a local surrogate model. For a given instance $x$, we draw a set $\mathcal{V}_x$ of 1000 instances uniformly from an $l_2$-ball of radius $r_{\text{fid}}$ centered on $x$. The local fidelity of the surrogate $\theta_x$ is measured as:

$$\text{LocalFid}(\theta_x) = \frac{1}{|\mathcal{V}_x|} \sum_{x' \in \mathcal{V}_x} \|f(x') - \mathcal{C}_{\theta_x}(x')\|_2,$$

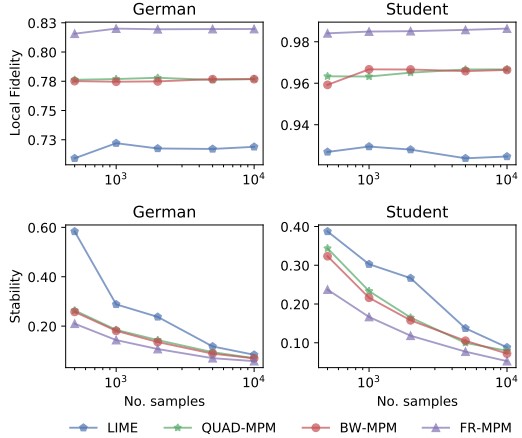

Figure 4: Benchmarks of fidelity and stability on the German and Student dataset. Higher local fidelity and lower stability are better.

where $f(\cdot)$ is the original classifier and $\mathcal{C}_\theta(\cdot)$ is the linear surrogate classifier. Basically, LocalFid measures the fraction of instances where the output class of $f$ and $\mathcal{C}_{\theta_x}$ agree. Here, we set $r_{\text{fid}}$ to 10% of the maximum distance between instances in the training data. Note that $\mathcal{V}_x$ is for evaluation only and independent from the perturbation samples used to train the local surrogate.

To generate the MPM's surrogates, we choose 10 nearest counterfactuals of $x_0$ in training data to find $x_b$. We set the perturbation radius $r_p$ to 5% of the maximum distance between instances in the training data and set $\rho_{+1} = 0$, $\rho_{-1} = 1.0$. For LIME, we use the default parameters recommended in the LIME source code and return $\theta = (w, b - 0.5)$ as the LIME's surrogate, similar to Laugel et al. (2018). We vary the number of perturbation samples in a range of $[500, 10000]$ to measure the fidelity and sensitivity of constructed surrogates under low sampling sizes. The results on German and Student datasets are shown in Figure 4. The results show the superiority of MPM's surrogates compared to LIME in both local fidelity and stability metrics. Meanwhile, FR-MPM provides higher-fidelity surrogates compared to QUAD-MPM and BW-MPM.

## 6.2 ROBUSTNESS OF RECOURSES

We now study the robustness to model shifts of recourses and its trade-off against the recourse cost.

**Metrics.** We use an $\ell_1$-distance as the *cost* function. We define the *current validity* as the validity of the generated recourses with respect to the given classifier. To measure the validity of recourses under

the model shifts, we sample $80\%$ instances of the shifted data $D_2$ 100 times to train 100 'future' MLP classifiers. We then report the *future validity* of recourse as the fraction of the future classifiers with respect to which the recourse is valid.

**Baselines.** The experiment in Section 6.1 suggests that the Fisher-Rao model can be used to represent the class of covariance-robust MPM. We use FR-MPM as the linear surrogate $\theta^{\mathbb{F}} = (w^{\mathbb{F}}, b^{\mathbb{F}})$ and a simple projection to generate recourse by finding $x_r = \arg\min\{\|x - x_0\|_1 : x^\top w^{\mathbb{F}} + b^{\mathbb{F}} \geq 0\}$.

We compare the above method, namely FR-MPM-PROJ, against strong baselines, including Wachter (Wachter et al., 2017) and ROAR (Upadhyay et al., 2021) using LIME (Ribeiro et al., 2016), CLIME (Agarwal et al., 2021), and LIMELS (Laugel et al., 2018)[2] as surrogates. Comparisons with ROAR using the vanilla MPM and SVM surrogates are in Appendix A.2.

**Cost-validity trade-off.** We fix the number of perturbation samples to 1000 and vary the ambiguity size ($\rho_{+1} = 0$, $\rho_{-1} \in [0, 10]$ for FR-MPM-PROJ and $\delta_{\max} \in [0, 0.2]$ for the uncertainty size of ROAR). We then plot the Pareto frontiers of the cost-validity trade-off in Figure 5. Generally, increasing the ambiguity size will increase both the current and future validity of recourses, but induces a sacrifice in the cost. This result is consistent with the analysis in Rawal et al. (2020). However, the frontiers of FR-MPM-PROJ dominate the frontiers of LIME-ROAR, CLIME-ROAR, and LIMELS-ROAR. Note that we use CARLA's implementation with default parameters for Wachter and its higher cost compared to the linear surrogate's methods on the Student dataset is consistent

Table 1: Performance of competing algorithms on real datasets. For the current and future validity, higher is better. For the cost, lower is better. Bold indicates the best performance.

| Method | German credit | | |
| --- | --- | --- | --- |
| | *Cost* | *Cur Validity* | *Fut Validity* |
| Wachter | **0.42** $\pm$ 0.02 | **1.00** $\pm$ 0.00 | 0.52 $\pm$ 0.03 |
| LIME-PROJ | 0.45 $\pm$ 0.10 | 0.35 $\pm$ 0.11 | 0.51 $\pm$ 0.08 |
| LIME-ROAR | 1.09 $\pm$ 0.19 | 0.66 $\pm$ 0.15 | 0.64 $\pm$ 0.08 |
| CLIME-ROAR | 1.40 $\pm$ 0.45 | 0.76 $\pm$ 0.17 | 0.72 $\pm$ 0.11 |
| LIMELS-ROAR | 1.29 $\pm$ 0.07 | 0.88 $\pm$ 0.03 | **0.79** $\pm$ 0.02 |
| QUAD-MPM-PROJ | 0.90 $\pm$ 0.02 | 0.89 $\pm$ 0.05 | 0.76 $\pm$ 0.03 |
| BW-MPM-PROJ | 1.03 $\pm$ 0.02 | 0.93 $\pm$ 0.04 | 0.78 $\pm$ 0.01 |
| FR-MPM-PROJ | 1.01 $\pm$ 0.02 | 0.94 $\pm$ 0.03 | 0.78 $\pm$ 0.01 |

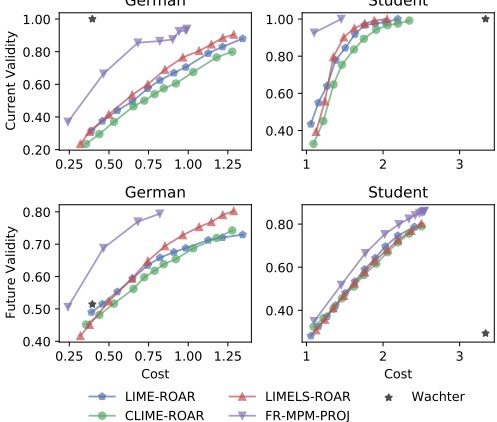

Figure 5: Pareto frontier of the cost-validity trade-off on the German and Student datasets.

with results in Upadhyay et al. (2021) and Pawelczyk et al. (2021). Table 1 shows the performance of FR-MPM-PROJ at radii $\rho_{+1} = 0$, $\rho_{-1} = 10$ and of ROAR-related methods at uncertainty size $\delta_{\max} = 0.2$. Here, the FR-MPM-PROJ has similar future validity compared to LIMELS-ROAR, but at a lower cost and higher current validity. Further results and discussion are in Appendix A.2.

**Actionability.** By introducing robustness into the linear surrogate model, our method can use AR (Ustun et al., 2019) as a recourse search mechanism to promote flexible actionability constraints. In Table 2, we compare the AR performance using different surrogate models (LIME, CLIME, LIMELS, SVM, MPMs). We consider the recourse under some actionability constraints such as immutable race, gender, or non-decrease age (see Appendix A.1 for more details on the actionability constraints). The result shows that AR

Table 2: Performance of AR using different local surrogate models.

| Method | German credit | | |
| --- | --- | --- | --- |
| | *Cost* | *Cur Validity* | *Fut Validity* |
| LIME-AR | 0.44 $\pm$ 0.08 | 0.15 $\pm$ 0.05 | 0.40 $\pm$ 0.07 |
| CLIME-AR | 0.73 $\pm$ 0.53 | 0.27 $\pm$ 0.12 | 0.45 $\pm$ 0.10 |
| LIMELS-AR | 0.36 $\pm$ 0.11 | 0.24 $\pm$ 0.09 | 0.44 $\pm$ 0.08 |
| SVM-AR | 0.39 $\pm$ 0.10 | 0.21 $\pm$ 0.08 | 0.42 $\pm$ 0.08 |
| MPM-AR | **0.29** $\pm$ 0.06 | 0.19 $\pm$ 0.07 | 0.42 $\pm$ 0.06 |
| QUAD-MPM-AR | 1.25 $\pm$ 0.09 | 0.77 $\pm$ 0.05 | 0.73 $\pm$ 0.03 |
| BW-MPM-AR | 1.86 $\pm$ 0.08 | 0.79 $\pm$ 0.03 | **0.74** $\pm$ 0.03 |
| FR-MPM-AR | 1.92 $\pm$ 0.10 | **0.80** $\pm$ 0.05 | **0.74** $\pm$ 0.03 |

using FR-MPM as the linear surrogate increases both the current and future validity substantially compared to other surrogates. The results for other datasets are presented in Appendix A.2

---

[2]LIMELS uses the same boundary sampling algorithm as the FR-MPM but trains a ridge regression instead.

## REPRODUCIBILITY STATEMENT

In order to foster reproducibility, we have released all source code and scripts used to replicate our experimental results at `https://anonymous.4open.science/r/mpm-recourse`. The repository includes source code, datasets, configurations, and instructions; thus one could reproduce our results with several commands.

We use the original authors' implementations of for LIME[3] (Ribeiro et al., 2016) and AR[4] (Ustun et al., 2019). We use a well-known CARLA's implementation[5] for Wachter (Wachter et al., 2017). Since we cannot find the open-source code for CLIME (Agarwal et al., 2021), LIMELS (Laugel et al., 2018), and ROAR (Upadhyay et al., 2021), we implement according to their papers. CLIME and LIMELS are adapted from LIME's source code while ROAR is adapted from CARLA's source code for Wachter.

The hyperparameter configurations for our methods and other baseline are clearly stated in Section A, Appendix A.1 and Appendix A.2 and also stored in the repository. The surrogates sharing the same local sampler will be ensured to have the same random seed, therefore, have the same synthesized samples. The hyperparameters that affect baselines' performance such as $\lambda$ and the probabilistic threshold of Wachter and ROAR will also be studied in Appendix A.2.

The remaining proofs and theoretical claims are provided in Appendix B.

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

# A  EXPERIMENTS

## A.1  EXPERIMENTAL DETAILS

**Classifier.** We use a three-layer MLP with 20, 50, and 20 nodes and a ReLU activation in each consecutive layer. We use the sigmoid function in the last layer to produce probabilities. To train this classifier, we use the binary cross-entropy, solved using the Adam optimizer and 1000 epochs.

**Datasets.** We provide more details about synthetic and real-world datasets. For synthetic data, we generate 2-dimensional data by sampling instances uniformly in a rectangle $x = (x_1, x_2) \in [-2, 4] \times [-2, 7]$. Each sample is labeled using the following function:

$$f(x) = \begin{cases} 1 & \text{if} \quad x_2 \geq 1 + x_1 + 2x_1^2 + x_1^3 - x_1^4 + \varepsilon, \\ 0 & \text{otherwise}, \end{cases}$$

where $\varepsilon$ is a random noise. We generate a present data set $D_1$ with $\varepsilon = 0$ and a shifted data set $D_2$ with $\varepsilon \sim \mathcal{N}(0, 1)$. The decision boundary of the MLP classifier for current synthetic data is illustrated in Figure 6.

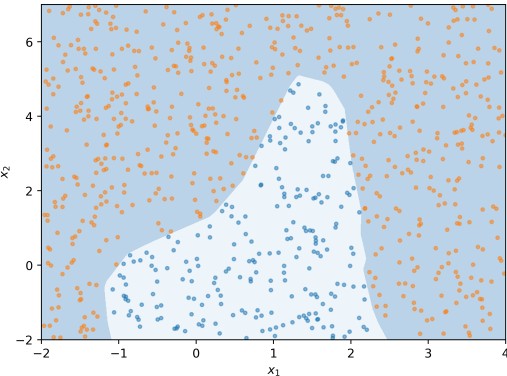

Figure 6: An illustration of MLP's decision boundary for the synthetic data.

The detail of three real-world datasets are listed below:

i *German Credit* (Dua & Graff, 2017). The dataset contains the information (e.g. age, gender, financial status,...) of 1000 customers who took a loan from a bank. The classification task is to determine the risk (good or bad) of an individual. There is another version of this dataset regarding corrections of coding error (Groemping, 2019). We use the corrected version of this dataset as shifted data to capture the correction shift. The features we used in this dataset include 'duration', 'amount', 'personal_status_sex', and 'age'. When considering actionability constraints (Section 6.2), we set 'personal_status_sex' as immutable and 'age' to be non-decrease.

ii *Small Bussiness Administration (SBA)* (Li et al., 2018). This data includes 2,102 observations with historical data of small business loan approvals from 1987 to 2014. We divide this dataset into two datasets (one is instances from 1989 - 2006 and one is instances from 2006 - 2014) to capture temporal shifts. We use the following features: selected, 'Term', 'NoEmp', 'CreateJob', 'RetainedJob', 'UrbanRural', 'ChgOffPrinGr', 'GrAppv', 'SBA_Appv', 'New', 'RealEstate', 'Portion', 'Recession'. When considering actionability constraints, we set 'UrbanRural' as immutable.

iii *Student performance* (Cortez & Silva, 2008). This data includes the performance records of 649 students in two schools: Gabriel Pereira (GP) and Mousinho da Silveira (MS). The classification task is to determine if their final score is above average or not. We split this dataset into two sets in two schools to capture geospatial shifts. The features we used are: 'age', 'Medu', 'Fedu', 'studytime', 'famsup', 'higher', 'internet', 'romantic', 'freetime', 'goout', 'health', 'absences', 'G1', 'G2'. When considering actionability constraints, we set 'romantic' as immutable and 'age' to be non-decreased.

For categorical features, we use one-hot encoding to convert them to binary features, similar to Mothilal et al. (2020). We also normalize continuous features to zero mean and unit variance before training the classifier. The performance of the classifier is reported in Table 3.

Table 3: Accuracy and AUC results of the classifiers on the synthetic and three real-world datasets.

| Classifier | Dataset | Present data $D_1$ | | Shift data $D_2$ | |
|---|---|---|---|---|---|
| | | Accuracy | AUC | Accuracy | AUC |
| MLP | Synthetic data | $0.99 \pm 0.00$ | $1.00 \pm 0.00$ | $0.94 \pm 0.01$ | $0.99 \pm 0.01$ |
| | German credit | $0.67 \pm 0.02$ | $0.60 \pm 0.03$ | $0.66 \pm 0.23$ | $0.60 \pm 0.04$ |
| | SBA | $0.96 \pm 0.00$ | $0.99 \pm 0.00$ | $0.98 \pm 0.01$ | $0.96 \pm 0.01$ |
| | Student | $0.86 \pm 0.02$ | $0.93 \pm 0.01$ | $0.91 \pm 0.04$ | $0.97 \pm 0.02$ |

## A.2 ADDITIONAL EXPERIMENTAL RESULTS

**Local fidelity and stability experiments.** Here, we provide benchmarks of local fidelity and stability (Section 6.1) on the synthetic and SBA datasets in Figure 7.

We also run with a different setting for the stability and local fidelity metric to assess if the results are sensitive to the parameter choices. Specifically, we sample 10 neighbors in the distribution $\mathcal{N}(x, 0.0001I)$ instead of $\mathcal{N}(x, 0.001I)$ to measure the stability. Meanwhile, we set $r_{fid}$ to 20% and the radius $r_p$ to 10% of the maximum distance between instances in the training data. The result is shown in Figure 8.

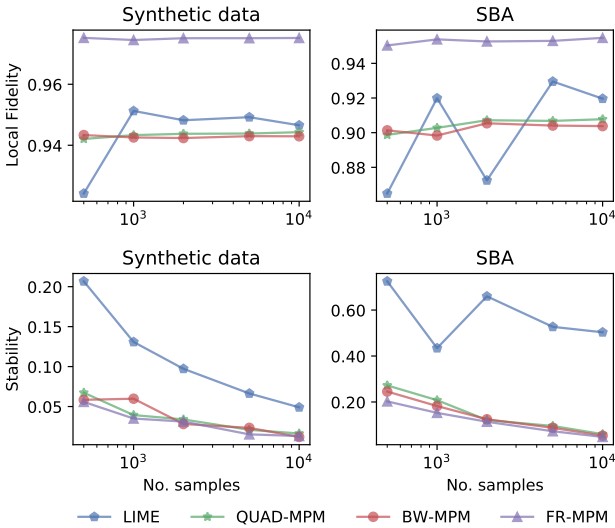

Figure 7: Benchmarks of the local fidelity and stability on synthetic and SBA dataset.

Table 4: Performance of competing algorithms on synthetic, SBA, and Student datasets.

| Method | Synthetic data | | | SBA | | | Student performance | | |
|---|---|---|---|---|---|---|---|---|---|
| | Cost | Cur Validity | Fut Validity | Cost | Cur Validity | Fut Validity | Cost | Cur Validity | Fut Validity |
| Wachter | **1.01** ± 0.03 | **1.00** ± 0.00 | 0.63 ± 0.04 | 1.74 ± 0.26 | 0.98 ± 0.01 | 0.13 ± 0.06 | 3.33 ± 0.11 | **0.95** ± 0.03 | 0.29 ± 0.02 |
| LIME-PROJ | 2.13 ± 0.08 | 0.77 ± 0.01 | 0.76 ± 0.01 | **0.81** ± 0.06 | 0.97 ± 0.01 | 0.79 ± 0.03 | **1.07** ± 0.13 | 0.78 ± 0.04 | 0.37 ± 0.07 |
| LIME-ROAR | 2.59 ± 0.09 | 0.90 ± 0.01 | 0.89 ± 0.00 | 1.83 ± 0.16 | **1.00** ± 0.00 | 0.87 ± 0.04 | 2.41 ± 0.31 | **0.95** ± 0.03 | 0.74 ± 0.11 |
| CLIME-ROAR | 1.80 ± 0.07 | 0.81 ± 0.01 | 0.81 ± 0.01 | 1.50 ± 0.53 | 0.97 ± 0.04 | 0.78 ± 0.13 | 3.07 ± 0.75 | 0.95 ± 0.04 | 0.81 ± 0.11 |
| LIMELS-ROAR | 1.30 ± 0.01 | **1.00** ± 0.00 | 0.98 ± 0.01 | 1.63 ± 0.23 | **1.00** ± 0.00 | 0.88 ± 0.05 | 2.47 ± 0.29 | **0.95** ± 0.03 | 0.75 ± 0.09 |
| QUAD-MPM-PROJ | 1.09 ± 0.02 | 0.99 ± 0.01 | 0.98 ± 0.00 | 1.17 ± 0.11 | **1.00** ± 0.00 | 0.85 ± 0.05 | 1.46 ± 0.13 | 0.95 ± 0.03 | 0.54 ± 0.07 |
| BW-MPM-PROJ | 1.19 ± 0.02 | **1.00** ± 0.00 | 0.99 ± 0.00 | 1.67 ± 0.13 | **1.00** ± 0.00 | 0.95 ± 0.02 | 2.13 ± 0.18 | **0.95** ± 0.03 | 0.76 ± 0.06 |
| FR-MPM-PROJ | 1.20 ± 0.02 | **1.00** ± 0.00 | **1.00** ± 0.00 | 1.83 ± 0.13 | **1.00** ± 0.00 | **0.97** ± 0.01 | 2.51 ± 0.21 | **0.95** ± 0.03 | **0.82** ± 0.05 |

**The cost-validity trade-off.** Here, we provide more detail about the settings of the baselines and the complementary results of the experiments in Section 6.2 in the main paper.

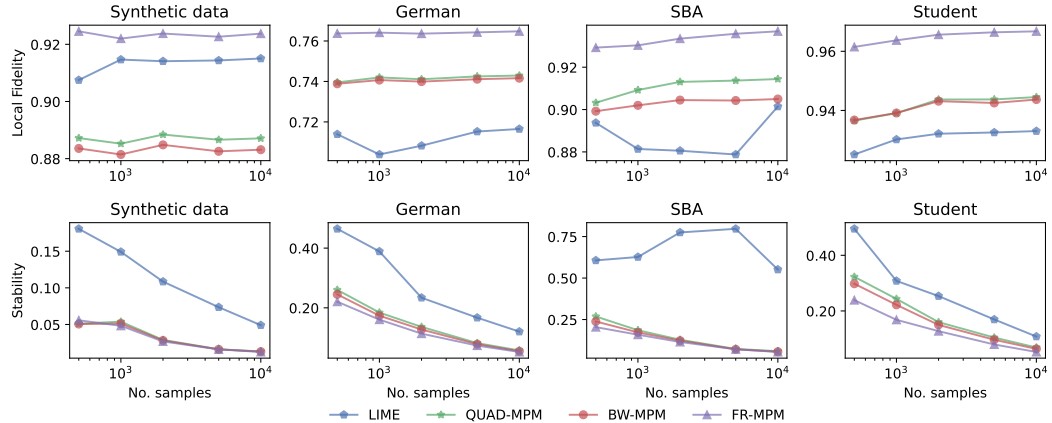

Figure 8: Benchmarks of the local fidelity and stability.

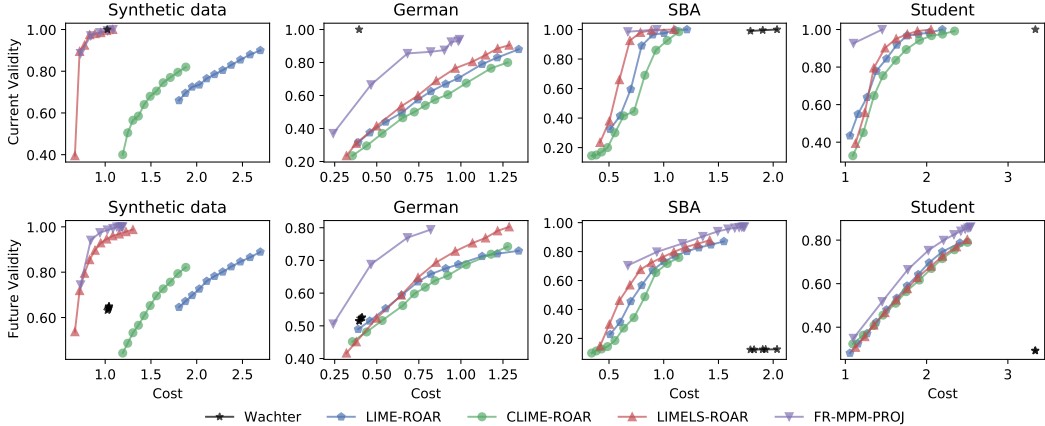

Figure 9: Pareto frontier of the cost-validity trade-off with the MLP classifier on the real-world datasets. For FR-MPM-PROJ, we vary the ambiguity size in a range of $\rho_- \in [0, 10]$. For the ROAR-related methods, we vary the uncertainty size in a range of $\delta_{\max} \in [0, 0.2]$. And for Wachter, we vary the hyperparameter $\lambda \in [0.05, 0.4]$.

For Wachter's implementation, we use CARLA's source code[6] (Pawelczyk et al., 2021), which employs an adaptive scheme to adjust the hyperparameter ($\lambda$) if no valid recourse is found. We adopt this implementation for ROAR and set the initial $\lambda$ to 0.1 as suggested in (Upadhyay et al., 2021).

Regarding the surrogate models, we use the open-source code with the default settings for LIME[7] (Ribeiro et al., 2016). We adapt this source code accordingly for CLIME's implementation (Agarwal et al., 2021). LIMELS, SVM, and MPM-related surrogates use the same boundary sampling procedure (with the same seed), in which we set the number of counterfactuals $k = 10$ and the number of perturbation samples is 1000.

Figure 9 shows the Pareto frontiers of the cost-validity trade-off on the synthetic and three real-world datasets. Table 4 shows the performance of FR-MPM-PROJ at radii $\rho_{+1} = 0, \rho_{-1} = 10$ and of ROAR-related methods at uncertainty size $\delta_{\max} = 0.2$ on synthetic, SBA, and Student datasets. The hyperparameter $\lambda$ is set to 0.1 for both Wachter and ROAR-related methods.

In FR-MPM-PROJ and ROAR-related methods, the trade-off between cost and validity can be observed when increasing the ambiguity size ($\rho_{-1}$ for FR-MPM and $\delta_{\max}$ for ROAR-related methods),

---

[6]https://github.com/carla-recourse/CARLA

[7]https://github.com/marcotcr/lime

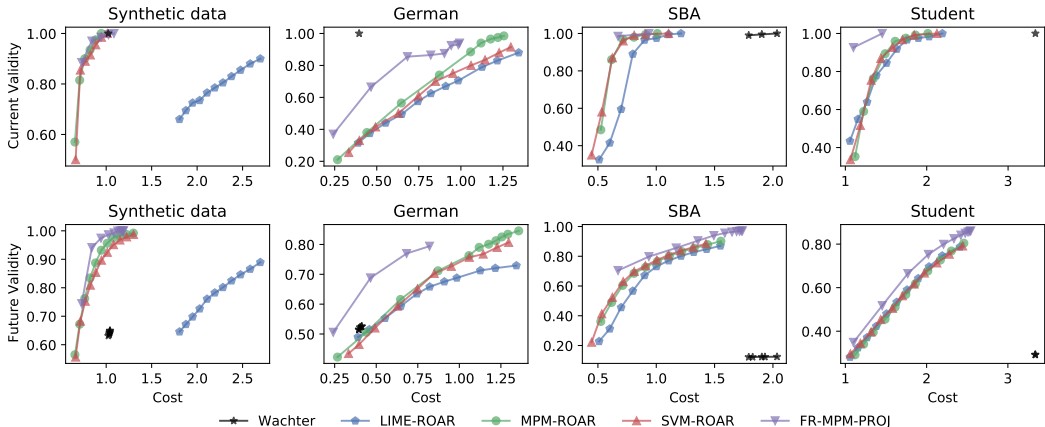

Figure 10: Pareto frontiers of our method compared with ROAR using vanilla MPM and SVM as the surrogate model. The recourses are generated with respect to the MLP classifier on synthetic and three real-world datasets.

Table 5: Performance of AR using different local surrogate models.

| Method | Synthetic data | | | SBA | | | Student performance | | |
|---|---|---|---|---|---|---|---|---|---|
| | Cost | Cur Validity | Fut Validity | Cost | Cur Validity | Fut Validity | Cost | Cur Validity | Fut Validity |
| LIME-AR | 2.06 ± 0.07 | 0.67 ± 0.02 | 0.67 ± 0.01 | 4.50 ± 3.48 | 0.10 ± 0.07 | 0.04 ± 0.03 | 3.38 ± 0.24 | 0.45 ± 0.07 | 0.42 ± 0.06 |
| CLIME-AR | 1.34 ± 0.05 | 0.41 ± 0.02 | 0.43 ± 0.01 | **4.10** ± 3.80 | 0.07 ± 0.07 | 0.01 ± 0.01 | 4.34 ± 1.24 | 0.63 ± 0.28 | 0.50 ± 0.15 |
| LIMELS-AR | 1.05 ± 0.05 | 0.62 ± 0.06 | 0.59 ± 0.04 | 4.97 ± 3.76 | 0.22 ± 0.15 | 0.06 ± 0.07 | 3.15 ± 0.16 | 0.49 ± 0.13 | 0.38 ± 0.03 |
| SVM-AR | **1.02** ± 0.06 | 0.64 ± 0.04 | 0.61 ± 0.02 | 6.06 ± 4.06 | 0.26 ± 0.07 | 0.03 ± 0.02 | 3.07 ± 0.23 | 0.43 ± 0.13 | 0.36 ± 0.03 |
| MPM-AR | 1.08 ± 0.04 | 0.68 ± 0.04 | 0.62 ± 0.03 | 5.06 ± 3.33 | 0.39 ± 0.18 | 0.08 ± 0.05 | **3.02** ± 0.19 | 0.40 ± 0.12 | 0.36 ± 0.02 |
| QUAD-MPM-AR | 1.65 ± 0.08 | 0.99 ± 0.01 | 0.98 ± 0.01 | 7.40 ± 3.36 | 0.97 ± 0.03 | 0.49 ± 0.19 | 5.60 ± 0.31 | 0.97 ± 0.04 | 0.69 ± 0.05 |
| BW-MPM-AR | 1.86 ± 0.09 | **1.00** ± 0.00 | 0.99 ± 0.01 | 9.66 ± 3.21 | 0.98 ± 0.04 | 0.62 ± 0.03 | 8.62 ± 0.38 | **1.00** ± 0.00 | 0.92 ± 0.05 |
| FR-MPM-AR | 2.03 ± 0.04 | **1.00** ± 0.00 | **0.99** ± 0.00 | 10.02 ± 3.09 | **1.00** ± 0.00 | **0.64** ± 0.03 | 9.63 ± 0.38 | **1.00** ± 0.00 | **0.95** ± 0.03 |

similar to the analysis in (Rawal et al., 2020) and the experimental results in (Upadhyay et al., 2021; Black et al., 2021). Generally, the Pareto frontiers of FR-MPM-PROJ dominate the frontiers of ROAR-related methods on all evaluated datasets. In other words, with the same cost (or validity), our method will provide recourses with a higher validity (or lower cost) compared to ROAR. Table 4 demonstrates that our method has similar validity but a much smaller cost than ROAR on Synthetic and German datasets. Meanwhile, our method achieves higher validity with reasonable cost on SBA and Student datasets.

Comparing other baselines, LIMELS-ROAR performs slightly better compared to LIME-ROAR and CLIME-ROAR. Wachter provides the recourses with high current validity but is vulnerable to model shifts, resulting in poor future validity. Wachter has the lowest cost for the synthetic and German datasets but a higher cost for the SBA and Student datasets compared to the linear surrogate-based methods. This is consistent with the results in (Upadhyay et al., 2021) since the objective function of Wachter might be non-convex when the classifier is an MLP network.

**Comparison with ROAR using the vanilla MPM and SVM as the surrogate model.** Here, we compare the FR-MPM-PROJ with ROAR using the vanilla MPM and SVM as the surrogate model. Both vanilla MPM and SVM use the same boundary sample procedure (with the same seed number) as FR-MPM. The settings are similar to the experiment in Section 6.2. The result shown in Figure 10 demonstrates the merits of our method.

**Actionability.** We provide the complementary result of the actionability experiment on the synthetic, SBA, and Student datasets in Table 5. Using AR with FR-MPM produces the recourses with higher value in both the current and future validity compared to AR using other surrogates.

**Ablation study.** We conduct an ablation study to understand the contribution of each stage in our method. Figure 10 shows the comparison of our method with ROAR using vanilla MPM and SVM as the local surrogate. Figure 11 shows the Pareto frontiers of FR-MPM-PROJ compared to its ablations by alternating the FR-MPM with other surrogates (LIME, MPM) or alternating the projection with

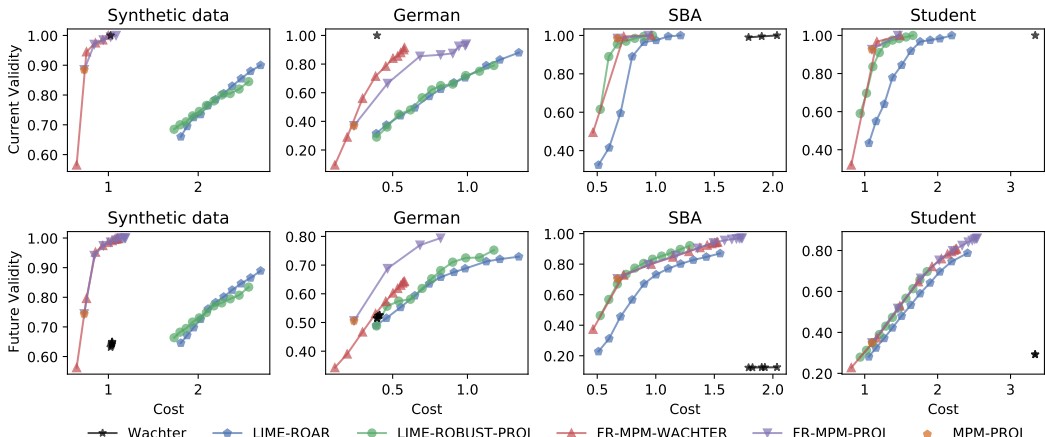

Figure 11: Ablation study: Pareto frontiers of FR-MPM-PROJ compared to its ablations by alternating the FR-MPM by common local surrogates. The recourses are generated with respect to the MLP classifier on synthetic and three real-world datasets.

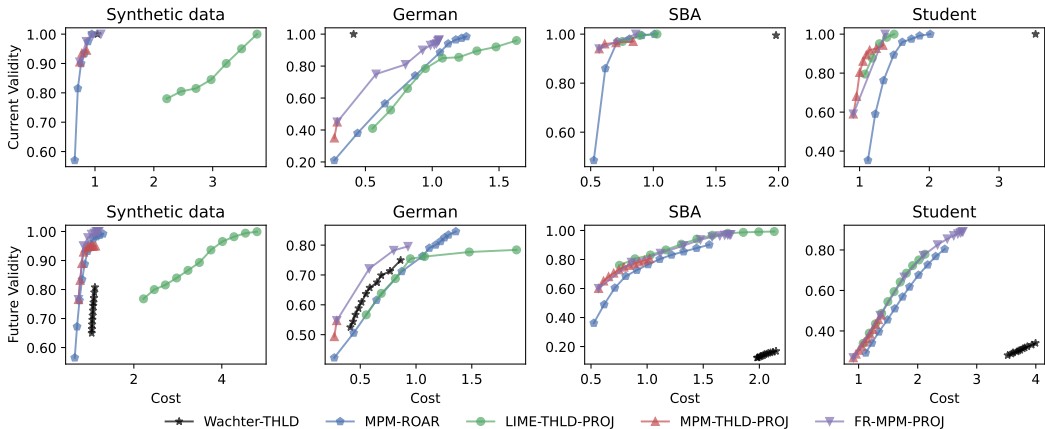

Figure 12: Comparison with the probabilistic shiftings.

Wachter. Particularly, we compare our method with LIME-ROBUST-PROJ, which uses LIME as the surrogate model and then solves the robustified projection:

$$x_r = \arg\min\{\|x - x_0\|_1 \; : \; x^\top w + b - \delta_{\max}\|x\|_2 \geq 0\},$$

where $(w, b)$ is the weight and bias of LIME's surrogate, $\delta_{\max}$ is similar to the uncertainty size of ROAR (Upadhyay et al., 2021). The recourses are generated with respect to the MLP classifier on synthetic and three real-world datasets. This result demonstrates the usefulness of the FR-MPM in promoting the generation of robust recourses. Note that, for $\rho_- = 0$ and $\rho_+ = 0$, the hyperplane of the FR-MPM classifier recovers the vanilla MPM's hyperplane.

**Comparison with the probabilistic shiftings.** One might attempt to increase the probabilistic threshold (usually set to $0.5$) at which a sample is considered 'favorable' to generate robust recourses. In Figure 12, we compare the proposed method with Wachter, LIME, and MPM with various probabilistic thresholds in the range $[0.5, 0.9]$. It can be seen that FR-MPM-PROJ consistently achieves the best performance compared to other baselines. Interestingly, Wachter improves its future validity significantly on the synthetic and German datasets as the threshold increases. However, our method still dominates Wachter in all datasets.

**Robust recourses with MPM's variants.** We compare FR-MPM-PROJ with QUAD-MPM-PROJ and BW-MPM-PROJ. The settings are similar in the cost-validity trade-off experiments in Section 6.2.

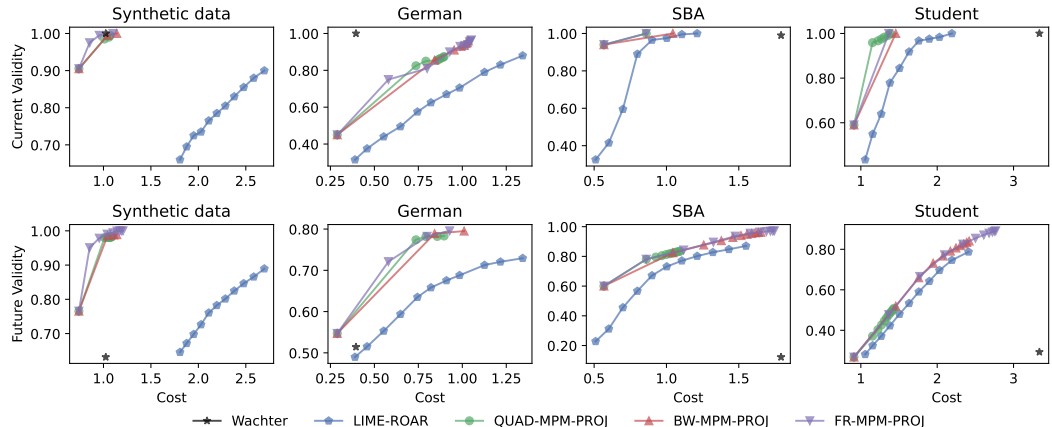

Figure 13: The comparison among MPM-related methods with different distances.

The number of samples is 1000. The results are shown in Figure 13. Generally, all three MPM-related methods perform better than LIME-ROAR and Wachter. However, FR-MPM-PROJ is better than QUAD-MPM-PROJ and BW-MPM-PROJ, especially in synthetic and Student datasets.

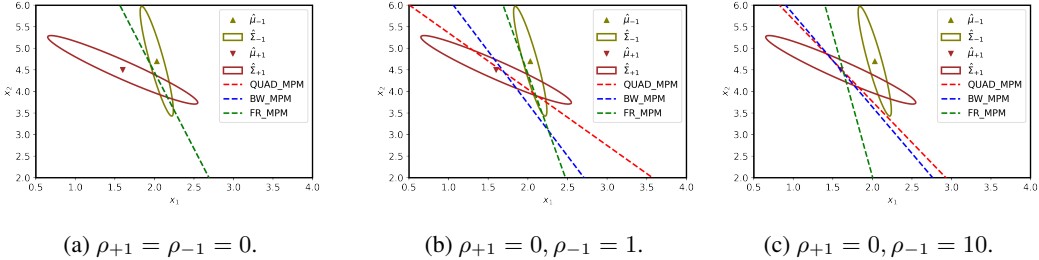

(a) $\rho_{+1} = \rho_{-1} = 0$.    (b) $\rho_{+1} = 0, \rho_{-1} = 1$.    (c) $\rho_{+1} = 0, \rho_{-1} = 10$.

Figure 14: Visualization of MPM's hyperplanes with Quadratic, Bures, and Fisher-Rao distances. When $\rho_{+1} = \rho_{-1} = 0$, all hyperplanes coincide and recover the vanilla MPM. All hyperplanes move towards the favorable class as the radius for the *un*favorable class $\rho_{-1}$ increases. At $\rho_{-1} = 10$ in Subfigure (c), the hyperplanes of Quadratic and Bures MPMs come close together which is distinct from the Fisher-Rao MPM's hyperplane. Notice that the Fisher-Rao MPM in Subfigure (c) tends to position in parallel to the major axis of the *un*favorable covariance matrix, which shows the dependence on $\widehat{\Sigma}_{-1}$, see Proposition 5.4. The Bures and Quad MPM hyperplanes in Subfigure (c) do not show any dependence on the covariance matrix, which aligns with the results in Proposition 4.4.

### A.3 COVARIANCE-ROBUST MPMs WITH DIFFERENT DIVERGENCES

In this section, we discuss the variants of covariance-robust MPMs with different distances and provide guidance for choosing the surrogate model in practice, especially at a low sample size.

Proposition 4.4 and Remark 4.5 showed that Quadratic MPM and Bures MPM coincide when one of the radii $\rho_y$ grows to infinity and they are independent of the covariance matrices $\widehat{\Sigma}_y$. Meanwhile, the asymptotic hyperplane of the Fisher-Rao MPM when $\rho_y \to \infty$ aligns with axes of the covariance matrices $\widehat{\Sigma}_y$ (see Proposition 5.4 and Figure 14). It suggests that the Fisher-Rao MPM is not a suitable surrogate at low sample sizes as it relies on the estimate of the covariance matrices. On the other hand, when the number of samples is sufficient to estimate the covariance matrices accurately, Fisher-Rao MPM would be better than Quadratic MPM and Bures MPM as it takes the geometry of the data into account when robustifying the surrogate.

We probe the performance of MPMs with different distances at low sample sizes to demonstrate our claim above.

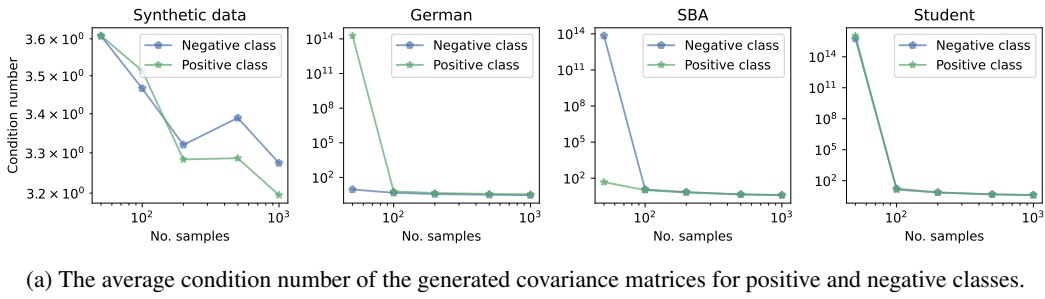

(a) The average condition number of the generated covariance matrices for positive and negative classes.

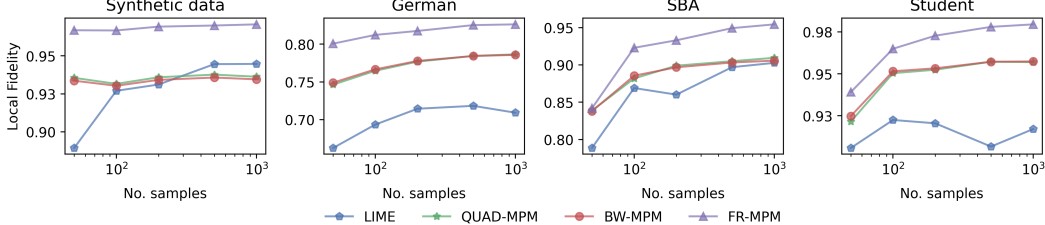

(b) Local fidelity of MPM variants at low sample sizes.

Figure 15: The comparison among MPM-related methods with different distances at low sample sizes.

**Local fidelity.** We probe the local fidelity at low sample sizes and plot the result in Figure 15. The experiment settings are similar to those in Section 6.1. The number of samples is set in the range of $[50, 1000]$. We also measure the average condition number of estimated covariance matrices for both positive and negative classes in Figure 15a. It can be seen that the covariance matrices are ill-conditioned at 50 samples on SBA and Student datasets. The fidelity of FR-MPM is just slightly better than QUAD-MPM and BW-MPM. When the number of samples increased, FR-MPM benefited the most, and the gap between FR-MPM and QUAD-MPM (or BW-MPM) became more significant. It supports our claim that FR-MPM would better approximate the decision boundary when the number of samples is sufficient for estimating the covariance matrices.

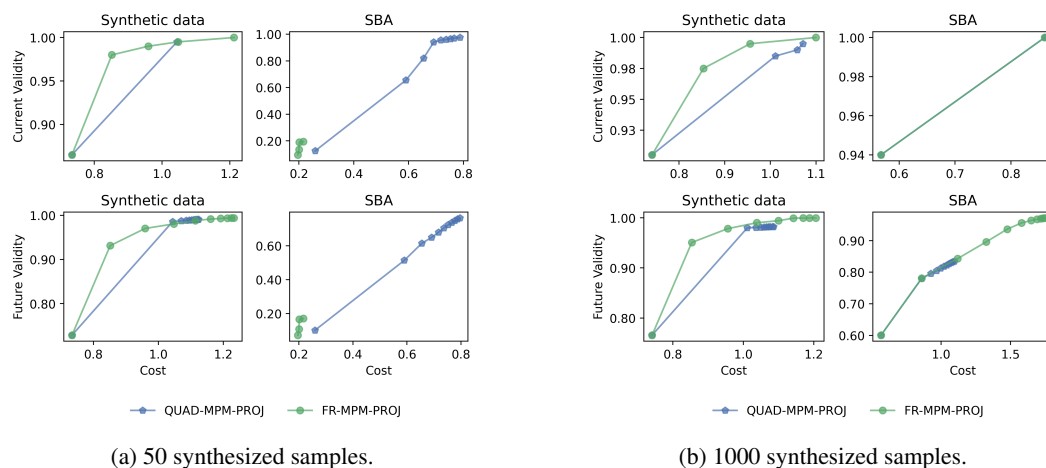

(a) 50 synthesized samples.     (b) 1000 synthesized samples.

Figure 16: The comparison of QUAD-MPM-PROJ and FR-MPM-PROJ at low sample sizes.

**Robust recourses.** We revisit the recourse generation with covariance-robust MPMs using quadratic distance (QUAD-MPM-PROJ) and Fisher-Rao distance (FR-MPM-PROJ), at which the surrogate is estimated with 50 and 1000 synthesized samples. We omit the comparison with Bures distance to ease the presentation as it behaves asymptotically like the MPM using the quadratic distance. The results are shown in Figure 16. The results showed that QUAD-MPM-PROJ would be better at a low

sample size. When increasing the number of samples, the recourses constructed with Fisher-Rao MPM exhibit a better cost-validity trade-off. This result is consistent with our previous observation in the local fidelity experiment.

# B PROOFS

## B.1 PROOFS OF SECTION 3

*Proof of Proposition 3.4.* The covariance-robust MPM problem

$$\min_{\theta \in \Theta} \max_{y \in \mathcal{Y}} \max_{\mathbb{P}_y \in \mathbb{U}_y^\varphi(\widehat{\mathbb{P}}_y)} \mathbb{P}_y(\mathcal{C}_\theta(X) \neq y)$$

is equivalent to

$$
\begin{aligned}
\min \quad & 1 - \alpha \\
\text{s.t.} \quad & \alpha \in \mathbb{R}_+, \ w \in \mathbb{R}^d \setminus \{0\}, \ b \in \mathbb{R} \\
& 1 - \alpha \geq \max_{y \in \mathcal{Y}} \max_{\mathbb{P}_y \in \mathbb{U}_y^\varphi(\widehat{\mathbb{P}}_y)} \mathbb{P}_y(y(w^\top X - b) \leq 0),
\end{aligned}
\tag{9}
$$

where we used the classification rule that $\mathcal{C}_\theta(x) = \text{sign}(w^\top x - b)$ if and only if $y(w^\top X - b) \geq 0$ and that the feasible set takes the form $\Theta \triangleq \{\theta = (w, b) \in \mathbb{R}^{d+1} : w \neq 0\}$. We claim that $w = 0$ is never optimal. To see this, take $w = 0$. Then,

$$\mathbb{P}_y(y(w^\top X - b) \leq 0) = \mathbb{P}_y(yb \geq 0),$$

which is independent of the random variable $X$ and is either 0 or 1 no matter what $b$ we choose. Therefore,

$$\max_{y \in \mathcal{Y}} \max_{\mathbb{P}_y \in \mathbb{U}_y^\varphi(\widehat{\mathbb{P}}_y)} \mathbb{P}_y(y(w^\top X - b) \leq 0) = 1,$$

and hence $\alpha = 0$, which is never optimal. So, the domain of $w$ can be relaxed from $\mathbb{R}^d \setminus \{0\}$ to $\mathbb{R}^d$. Problem (9) can then be further re-written as

$$
\begin{aligned}
\max \quad & \alpha \\
\text{s.t.} \quad & \alpha \in \mathbb{R}_+, \ w \in \mathbb{R}^d, \ b \in \mathbb{R} \\
& 1 - \alpha \geq \max_{\mathbb{P}_y \in \mathbb{U}_y^\varphi(\widehat{\mathbb{P}}_y)} \mathbb{P}_y(y(w^\top X - b) \leq 0) \qquad \forall y \in \mathcal{Y}.
\end{aligned}
\tag{10}
$$

Notice that here, equivalency means the optimal solution $(w^\star, b^\star)$ of (10) will constitute the optimal solution $\theta^\star = (w^\star, b^\star)$ of the original min-max-max problem. Moreover, recall the definition of the ambiguity set

$$\mathbb{U}_y^\varphi(\widehat{\mathbb{P}}_y) = \{\mathbb{P}_y : \ \mathbb{P}_y \sim (\widehat{\mu}_y, \Sigma_y), \ \varphi(\Sigma_y \parallel \widehat{\Sigma}_y) \leq \rho_y\},$$

where $\mathbb{P}_y \sim (\widehat{\mu}_y, \Sigma_y)$ means that the the distribution $\mathbb{P}_y$ has mean $\widehat{\mu}_y$ and covariance $\Sigma_y$. In other words, each element $\mathbb{P}_y$ in the ambiguity $\mathbb{U}_y^\varphi(\widehat{\mathbb{P}}_y)$ is determined by first choosing a covariance matrix $\Sigma_y$ satisfying the divergence constraint $\varphi(\Sigma_y \parallel \widehat{\Sigma}_y) \leq \rho_y$ and then picking a distribution $\mathbb{P}_y$ having mean $\widehat{\mu}_y$ and covariance $\Sigma_y$. Therefore, the worst-case probability admits a two-layer decomposition

$$\max_{\mathbb{P}_y \in \mathbb{U}_y^\varphi(\widehat{\mathbb{P}}_y)} \mathbb{P}_y(y(w^\top X - b) \leq 0) = \max_{\Sigma_y \in \mathbb{S}_+^d : \varphi(\Sigma_y \| \widehat{\Sigma}_y) \leq \rho_y} \max_{\mathbb{P}_y \sim (\widehat{\mu}_y, \Sigma_y)} \mathbb{P}_y(y(w^\top X - b) \leq 0). \tag{11}$$

Using Lanckriet et al. (2001, Equation (6)), the inner maximum value is given by

$$\max_{\mathbb{P}_y \sim (\widehat{\mu}_y, \Sigma_y)} \mathbb{P}_y(y(w^\top X - b) \leq 0) = \frac{1}{1 + \frac{(b - w^\top \widehat{\mu}_y)^2}{w^\top \Sigma_y w}}.$$

Combining the last two equalities, we can express the constraint in problem (10) as

$$1 - \alpha \geq \max_{\mathbb{P}_y \in \mathbb{U}_y^\varphi(\widehat{\mathbb{P}}_y)} \mathbb{P}_y(y(w^\top X - b) \leq 0) = \max_{\Sigma_y \in \mathbb{S}_+^d : \varphi(\Sigma_y \| \widehat{\Sigma}_y) \leq \rho_y} \frac{1}{1 + \frac{(b - w^\top \widehat{\mu}_y)^2}{w^\top \Sigma_y w}},$$

which upon rearranging, becomes

$$\frac{\alpha}{1 - \alpha} \max_{\Sigma_y \in \mathbb{S}_+^d : \varphi(\Sigma_y \| \widehat{\Sigma}_y) \leq \rho_y} w^\top \Sigma_y w \leq (b - w^\top \widehat{\mu}_y)^2.$$

Using the same argument as in Lanckriet et al. (2001) (see equation (4) and the discussions following it in Lanckriet et al. (2001)), we could show that the optimal $\theta = (w, b)$ must classify $\widehat{\mu}_y$ correctly, *i.e.*,

$$y = \text{sign}(w^\top \widehat{\mu}_y - b),$$

which yields that

$$\sqrt{\frac{\alpha}{1 - \alpha}} \max_{\Sigma_y \in \mathbb{S}_+^d : \varphi(\Sigma_y \| \widehat{\Sigma}_y) \le \rho_y} \sqrt{w^\top \Sigma_y w} \le y(w^\top \widehat{\mu}_y - b).$$

As a consequence, problem (10) is equivalent to

$$\begin{aligned}
\max \quad & \alpha \\
\text{s.t.} \quad & \alpha \in \mathbb{R}_+, \ w \in \mathbb{R}^d, \ b \in \mathbb{R} \\
& y(w^\top \widehat{\mu}_y - b) \ge \sqrt{\frac{\alpha}{1-\alpha}} \max_{\Sigma_y \in \mathbb{S}_+^d : \varphi(\Sigma_y \| \widehat{\Sigma}_y) \le \rho_y} \sqrt{w^\top \Sigma_y w} \qquad \forall y \in \mathcal{Y}.
\end{aligned}$$

Using that $\tau_y^\varphi(w) = \max_{\Sigma_y \in \mathbb{S}_+^d : \varphi(\Sigma_y \| \widehat{\Sigma}_y) \le \rho_y} \sqrt{w^\top \Sigma_y w}$ and that $\alpha \mapsto \sqrt{\frac{\alpha}{1-\alpha}}$ is monotone increasing, the above problem is further equivalent to

$$\begin{aligned}
\max \quad & \kappa \\
\text{s.t.} \quad & \kappa \in \mathbb{R}_+, \ w \in \mathbb{R}^d, \ b \in \mathbb{R} \\
& y(w^\top \widehat{\mu}_y - b) \ge \kappa \, \tau_y^\varphi(w) \qquad \forall y \in \mathcal{Y}.
\end{aligned} \tag{12}$$

From the constraints, we get

$$w^\top \widehat{\mu}_{+1} - \kappa \, \tau_{+1}^\varphi(w) \ge b \ge w^\top \widehat{\mu}_{-1} + \kappa \, \tau_{-1}^\varphi(w) \tag{13}$$

So, we can eliminate the variable $b$ and reduce problem (12) to

$$\begin{aligned}
\max \quad & \kappa \\
\text{s.t.} \quad & \kappa \in \mathbb{R}_+, \ w \in \mathbb{R}^d \\
& w^\top \widehat{\mu}_{+1} - \kappa \, \tau_{+1}^\varphi(w) \ge w^\top \widehat{\mu}_{-1} + \kappa \, \tau_{-1}^\varphi(w).
\end{aligned} \tag{14}$$

The inequality constraint in problem (14) is equivalent to

$$\kappa \le \frac{\sum_{y \in \mathcal{Y}} y \, w^\top \widehat{\mu}_y}{\sum_{y \in \mathcal{Y}} \tau_y^\varphi(w)}. \tag{15}$$

Thus, we can eliminate the variable $\kappa$ and rewrite problem (14) as

$$\min_{w \in \mathbb{R}^d} \frac{\sum_{y \in \mathcal{Y}} \tau_y^\varphi(w)}{\sum_{y \in \mathcal{Y}} y \, w^\top \widehat{\mu}_y}.$$

Using the definition of $\tau_y^\varphi(w)$, we can see that the above problem is homogeneous in $w$, which implies that we could further re-write it as

$$\begin{aligned}
\min \quad & \sum_{y \in \mathcal{Y}} \tau_y^\varphi(w) \\
\text{s.t.} \quad & \sum_{y \in \mathcal{Y}} y \, w^\top \widehat{\mu}_y = 1.
\end{aligned}$$

Finally, note that from (13) and (15), at optimality, we have

$$\kappa = \frac{\sum_{y \in \mathcal{Y}} y \, w^\top \widehat{\mu}_y}{\sum_{y \in \mathcal{Y}} \tau_y^\varphi(w)} = \frac{1}{\sum_{y \in \mathcal{Y}} \tau_y^\varphi(w)},$$

and

$$b = w^\top \widehat{\mu}_{+1} - \kappa \, \tau_{+1}^\varphi(w) = w^\top \widehat{\mu}_{-1} + \kappa \, \tau_{-1}^\varphi(w).$$

This completes the proof. $\qquad\square$

*Proof of Proposition 3.5.* First, following exactly the same arguments as in the proof of Proposition 3.4, we see that problem (6) is equivalent to

$$
\begin{aligned}
\max \quad & \alpha \\
\text{s.t.} \quad & \alpha \in \mathbb{R}_+, \ w \in \mathbb{R}^d, \ b \in \mathbb{R} \\
& 1 - \alpha \geq \max_{\mathbb{P}_y \in \mathcal{U}_y^{\mathcal{N}}(\widehat{\mathbb{P}}_y)} \mathbb{P}_y(y(w^\top X - b) \leq 0) \qquad \forall y \in \mathcal{Y}.
\end{aligned}
\tag{16}
$$

In the proof of Proposition 3.4, we handle the maximum in the constraint by decomposing it into two layers of maximization problems (see (11)). However, because of the Gaussian assumption, in this case, we have

$$
\max_{\mathbb{P}_y \in \mathcal{U}_y^{\mathcal{N}}(\widehat{\mathbb{P}}_y)} \mathbb{P}_y(y(w^\top X - b) \leq 0) = \max_{\substack{\Sigma_y \in \mathbb{S}_+^d : \varphi(\Sigma_y \| \widehat{\Sigma}_y) \leq \rho_y \\ \mathbb{P}_y \sim \mathcal{N}(\widehat{\mu}_y, \Sigma_y)}} \mathbb{P}_y(y(w^\top X - b) \leq 0).
$$

For each fixed $\Sigma_y$, using elementary probability theory, we could calculate the Gaussian probability explicitly:

$$
\mathbb{P}_y(y(w^\top X - b) \leq 0) = 1 - \Phi\left( \frac{y(w^\top \widehat{\mu}_y - b)}{\sqrt{w^\top \Sigma_y w}} \right),
$$

where $\Phi$ is the cumulative distribution function of the standard Gaussian random variable. Therefore, problem (16) can be re-written as

$$
\begin{aligned}
\max \quad & \alpha \\
\text{s.t.} \quad & \alpha \in \mathbb{R}_+, \ w \in \mathbb{R}^d, \ b \in \mathbb{R} \\
& \alpha \leq \max_{\Sigma_y \in \mathbb{S}_+^d : \varphi(\Sigma_y \| \widehat{\Sigma}_y) \leq \rho_y} \Phi\left( \frac{y(w^\top \widehat{\mu}_y - b)}{\sqrt{w^\top \Sigma_y w}} \right) \qquad \forall y \in \mathcal{Y}.
\end{aligned}
\tag{17}
$$

Using the monotonicity of $\Phi$, the constraints in problem (17) become

$$
y(w^\top \widehat{\mu}_y - b) \geq \Phi^{-1}(\alpha) \min_{\Sigma_y \in \mathbb{S}_+^d : \varphi(\Sigma_y \| \widehat{\Sigma}_y) \leq \rho_y} \sqrt{w^\top \Sigma_y w} \qquad \forall y \in \mathcal{Y}.
$$

Using that $\tau_y^\varphi(w) = \max_{\Sigma_y \in \mathbb{S}_+^d : \varphi(\Sigma_y \| \widehat{\Sigma}_y) \leq \rho_y} \sqrt{w^\top \Sigma_y w}$ and that $\Phi^{-1}(\alpha)$ is monotone increasing, problem (17) is further equivalent to

$$
\begin{aligned}
\max \quad & \kappa \\
\text{s.t.} \quad & \kappa \in \mathbb{R}_+, \ w \in \mathbb{R}^d, \ b \in \mathbb{R} \\
& y(w^\top \widehat{\mu}_y - b) \geq \kappa \, \tau_y^\varphi(w) \qquad \forall y \in \mathcal{Y}.
\end{aligned}
$$

which is the same problem as problem (12) in the proof of Proposition 3.4. Hence, problem (6) shares the same optimal solution as problem (3). This completes the proof. $\square$

## B.2 PROOFS OF SECTION 4

We first prove Proposition 4.3 to lay the foundation for the proof of Theorem 4.2.

*Proof of Proposition 4.3.* By Nguyen et al. (2021, Proposition 2.8), we have

$$
\tau_y^{\mathbb{B}}(w)^2 = \inf_{\gamma I \succ w w^\top} \gamma(\rho_y - \text{Tr}\left[ \widehat{\Sigma}_y \right]) + \gamma^2 \langle (\gamma I - w w^\top)^{-1}, \widehat{\Sigma}_y \rangle.
$$

Using the Sherman-Morrison formula (Bernstein, 2009, Corollary 2.8.8), we find

$$
(I - \frac{1}{\gamma} w w)^{-1} = I + \frac{w w^\top}{\gamma - \|w\|_2^2}.
$$

Notice that the constraint $\gamma I \succ w w^\top$ is equivalent to $\gamma > \|w\|_2^2$ by Schur complement. Thus, we have

$$
\tau_y^{\mathbb{B}}(w)^2 = \inf_{\gamma > \|w\|_2^2} \gamma \rho_y + \gamma \frac{w^\top \widehat{\Sigma}_y w}{\gamma - \|w\|_2^2}.
$$

The optimal $\gamma$ can be found by using calculus, which is given by

$$\gamma^\star = \|w\|_2^2 + \sqrt{\frac{w^\top \widehat{\Sigma}_y w \|w\|_2^2}{\rho_y}},$$

with the corresponding optimal value

$$\tau_y^{\mathbb{B}}(w)^2 = (\rho_y \|w\|_2 + \sqrt{w^\top \widehat{\Sigma}_y w})^2.$$

We thus have the necessary result. $\qquad\square$

We now prove Theorem 4.2.

*Proof of Theorem 4.2.* Using the Bures divergence $\mathbb{B}$, the optimization problem

$$\min_{w \in \mathcal{W}} \sum_{y \in \mathcal{Y}} \tau_y^{\mathbb{B}}(w)$$

becomes problem (7) by exploiting the analytical form of $\tau_y^{\mathbb{B}}(w)$ in Proposition 4.3. By invoking Proposition 3.4, we obtain the postulated results on the optimal solution $\theta^{\mathbb{B}}$ for the case of the Bures divergence. $\qquad\square$

*Proof of Proposition 4.4.* Note that problem (7) has a unique solution because the objective function is strictly convex and coercive. Moreover, the optimal solution of (7) coincides with the optimal solution $w^\star(\lambda)$ of the following second-order cone program

$$\min_{w \in \mathcal{W}} \frac{1}{\lambda} \sum_{y \in \mathcal{Y}} \sqrt{w^\top \widehat{\Sigma}_y w} + \|w\|_2,$$

where $\lambda = 1/\rho$. By a compactification of $\mathcal{W}$ and applying Berge's maximum theorem (Berge, 1963, pp. 115-116), the function $w^\star(\lambda)$ is continuous on a non-negative compact range of $\lambda$, and converges to $w^\star(0)$ as $\lambda \to 0$. The optimal solution $w^\star(0)$ coincides with the solution of

$$\min_{w \in \mathcal{W}} \|w\|_2, \tag{18}$$

which is the Euclidean projection of the origin onto the hyperplane $\mathcal{W}$. An elementary computation confirms that

$$w^\star(0) = \frac{\sum_{y \in \mathcal{Y}} y \widehat{\mu}_y}{\|\sum_{y \in \mathcal{Y}} y \widehat{\mu}_y\|_2^2}.$$

Letting $w_\infty^{\mathbb{B}} = w^\star(0)$ completes the proof. $\qquad\square$

### B.3 PROOFS OF SECTION 5

We first provide the proof of Proposition 5.3.

*Proof of Proposition 5.3.* Notice that

$$\tau_y^{\mathbb{F}}(w)^2 = \begin{cases} \max & w^\top \Sigma_y w \\ \text{s.t.} & \|\log(\widehat{\Sigma}_y^{-\frac{1}{2}} \Sigma_y \widehat{\Sigma}_y^{-\frac{1}{2}})\|_F \leq \rho_y. \end{cases}$$

Using the transformation $Z_y \leftarrow \widehat{\Sigma}_y^{-\frac{1}{2}} \Sigma_y \widehat{\Sigma}_y^{-\frac{1}{2}}$, we have

$$\tau_y^{\mathbb{F}}(w)^2 = \max \left\{ v^\top Z_y v \ : \ \|\log Z_y\|_F \leq \rho_y \right\}$$

with $v = \widehat{\Sigma}^{\frac{1}{2}} w$. We now proceed to show that the above optimization problem admits the maximizer

$$Z_y^\star = U U^\top + \exp(\rho_y) \frac{v v^\top}{\|v\|_2^2},$$

where $U$ is an $d \times (d-1)$ orthonormal matrix whose columns are orthogonal to $v$. First, by Nguyen et al. (2019, Lemma C.1), the feasible region is compact. Since the objective function $v^\top Z_Y v$ is continuous in $Z_y$, an optimal solution $Z_y^\star$ exists. Next, we first claim that the constraint holds with equality at optimality. Suppose that $\|\log Z_y^\star\|_F < \rho_y$. Then, for some small $\delta > 0$, the matrix $Z_y^\star + \delta\, vv^\top$ is feasible due to the continuity of the constraint function $\|\log Z_y\|_F$ and has a strictly better objective value than the optimal solution $Z_y^\star$. This violates the optimality of $Z_y^\star$. Hence, $\|\log Z_y^\star\|_F = \rho_y$ for any optimal solution $Z_y^\star$, and the problem is equivalent to

$$
\begin{aligned}
\max \quad & v^\top Q \mathrm{Diag}(\lambda) Q^\top v \\
\mathrm{s.\,t.} \quad & \sum_{i=1}^d (\log \lambda_i)^2 = \rho_y^2, \\
& \lambda_1 \geq \cdots \geq \lambda_d > 0, \ Q \in \mathcal{O}(d),
\end{aligned}
$$

where $\mathcal{O}(d)$ is the set of $d \times d$ orthogonal matrices. For any orthogonal matrix $Q$, the objective function

$$
v^\top Q \mathrm{Diag}(\lambda) Q^\top v \leq \lambda_1 \|v\|_2^2,
$$

the right-hand side of which can be attained by setting

$$
Q = \left( \frac{v}{\|v\|_2}, \ U \right) \in \mathcal{O}(d).
$$

Therefore, our problem is further reduced to

$$
\begin{aligned}
\max \quad & \lambda_1 \|v\|_2^2 \\
\mathrm{s.\,t.} \quad & \sum_{i=1}^d (\log \lambda_i)^2 = \rho_y^2, \ \lambda_1 \geq \cdots \geq \lambda_d > 0.
\end{aligned}
$$

It is then easy to see that at optimality, the optimal $\lambda \in \mathbb{R}_{++}^d$ must satisfy $\lambda_2 = \cdots = \lambda_d = 1$ and $(\log \lambda_1)^2 = \rho_y^2$. Since $\lambda_1 \geq \lambda_2 = 1$, we have $\log \lambda_1 = \rho_y$ and hence $\lambda_1 = \exp(\rho_y)$. In other words,

$$
Z_y^\star = \left( \frac{v}{\|v\|_2}, \ U \right)
\begin{pmatrix} \exp(\rho_y) & & & \\ & 1 & & \\ & & \ddots & \\ & & & 1 \end{pmatrix}
\left( \frac{v}{\|v\|_2}, \ U \right)^\top
$$

$$
= UU^\top + \exp(\rho_y) \frac{vv^\top}{\|v\|_2^2}.
$$

The corresponding optimal value is

$$
\tau_y^{\mathbb{F}}(w)^2 = v^\top Z_y^\star v = \exp(\rho_y) \|v\|_2^2 = \exp(\rho_y)\, w^\top \widehat{\Sigma}_y w.
$$

This completes the proof. $\qquad\square$

We are now ready to prove Theorem 5.2.

*Proof of Theorem 5.2.* Using the Fisher-Rao divergence $\mathbb{F}$, the optimization problem

$$
\min_{w \in \mathcal{W}} \ \sum_{y \in \mathcal{Y}} \tau_y^{\mathbb{F}}(w)
$$

becomes problem (8) by exploiting the analytical form of $\tau_y^{\mathbb{F}}(w)$ in Proposition 5.3. By invoking Proposition 3.4, we obtain the postulated results on the optimal solution $\theta^{\mathbb{F}}$ for the case of the Fisher-Rao divergence. $\qquad\square$

*Proof of Proposition 5.4.* Note that problem (8) has a unique solution because the objective function is strictly convex and coercive. Also, the optimal solution of (8) coincides with the solution $w^\star(\lambda)$ of

$$
\min_{w \in \mathcal{W}} \ \sqrt{w^\top \widehat{\Sigma}_y w} + \lambda \sqrt{w^\top \widehat{\Sigma}_{-y} w},
$$

where $\lambda = \exp\left(\frac{\rho_{-y} - \rho_y}{2}\right)$. By a compactification of $\mathcal{W}$ and applying Berge's maximum theorem (Berge, 1963, pp. 115-116), the function $w^\star(\lambda)$ is continuous on a non-negative compact range of $\lambda$, and converges to $w^\star(0)$ as $\lambda \to 0$. The optimal solution $w^\star(0)$ coincides with the solution of

$$\min_{w \in \mathcal{W}} \sqrt{w^\top \widehat{\Sigma}_y w}.$$

Because the square-root function is monotonically increasing, $w^\star(0)$ also solves

$$\min_{w \in \mathcal{W}} w^\top \widehat{\Sigma}_y w,$$

which is a convex, quadratic program with a single linear constraint. If $a$ is defined as in the statement, then a convex optimization argument implies

$$w^\star(0) = \frac{1}{a^\top \widehat{\Sigma}_y^{-1} a} \widehat{\Sigma}_y^{-1} a,$$

which completes the proof. $\qquad\square$

## C  LOGDET MPM

In this appendix, we consider when $\varphi$ is the Log-Determinant (LogDet) divergence. The LogDet divergence is formally defined as follows.

**Definition C.1** (LogDet divergence). *Given two positive definite matrices $\Sigma, \widehat{\Sigma} \in \mathbb{S}_{++}^d$, the log-determinant divergence between them is*

$$\mathbb{D}(\Sigma \parallel \widehat{\Sigma}) = \mathrm{Tr}\left[\Sigma\widehat{\Sigma}^{-1}\right] - \log\det(\Sigma\widehat{\Sigma}^{-1}) - d.$$

It can be shown that $\mathbb{D}$ is a divergence because it is non-negative, and it vanishes to zero if and only if $\Sigma = \widehat{\Sigma}$. However, $\mathbb{D}$ is not symmetric, and in general we have $\mathbb{D}(\Sigma \parallel \widehat{\Sigma}) = \mathbb{D}(\widehat{\Sigma} \parallel \Sigma)$. The LogDet divergence $\mathbb{D}$ is related to the relative entropy: it is equal to the Kullback-Leibler divergence between two Gaussian distributions with the same mean vector and covariance matrices $\Sigma$ and $\widehat{\Sigma}$.

We now provide the form of the LogDet MPM problem.

**Theorem C.2** (LogDet MPM). *Suppose that $\varphi \equiv \mathbb{D}$. Let $w^{\mathbb{D}}$ be the optimal solution of the following second-order cone problem*

$$\min_{w \in \mathcal{W}} \sum_{y \in \mathcal{Y}} \sqrt{c_y} \sqrt{w^\top \widehat{\Sigma}_y w}, \tag{19}$$

*where $c_y = -W_{-1}(-\exp(-\rho_y - 1))$ and $W_{-1}$ is the Lambert-W function for the branch $-1$. Let $\kappa^{\mathbb{D}}$ and $b^{\mathbb{D}}$ be calculated as*

$$\kappa^{\mathbb{D}} = \frac{1}{\sum_{y \in \mathcal{Y}} \sqrt{c_y} \sqrt{(w^{\mathbb{D}})^\top \widehat{\Sigma}_y w^{\mathbb{D}}}},$$

$$b^{\mathbb{D}} = (w^{\mathbb{D}})^\top \widehat{\mu}_{+1} - \kappa^{\mathbb{D}} \sqrt{c_{+1}} \sqrt{(w^{\mathbb{D}})^\top \widehat{\Sigma}_{+1} w^{\mathbb{D}}},$$

*then $\theta^{\mathbb{D}} = (w^{\mathbb{D}}, b^{\mathbb{D}})$ is the optimal solution of the distributionally robust MPM problem* (3).

Theorem C.2 shows that the LogDet divergence induces a similar reweighting scheme as the Fisher-Rao MPM. The asymptotic analysis of the LogDet MPM follows similarly from the Fisher-Rao MPM and is omitted. The proof of Theorem C.2 follows trivially from the below result, which provides the analytical form of $\tau_y^{\mathbb{D}}(w)$.

**Proposition C.3** (LogDet divergence). *Suppose that $\varphi \equiv \mathbb{D}$, then for any $y \in \mathcal{Y}$, we have*

$$\tau_y^{\mathbb{D}}(w) = \sqrt{-W_{-1}(-\exp(-\rho_y - 1))} \sqrt{w^\top \widehat{\Sigma}_y w},$$

*where $W_{-1}$ is the Lambert-W function for the branch $-1$.*

*Proof of Proposition C.3.* By Le et al. (2021, Proposition 3.4), we have

$$\tau_y^{\mathrm{D}}(w)^2 = \inf_{\substack{\gamma > 0 \\ \gamma \widehat{\Sigma}_y^{-1} \succ ww^\top}} \gamma \rho_y - \gamma \log \det(I - \widehat{\Sigma}_y^{\frac{1}{2}} ww^\top \widehat{\Sigma}_y^{\frac{1}{2}} / \gamma).$$

Using the matrix determinant formula (Bernstein, 2009), we have

$$\det(I - \widehat{\Sigma}_y^{\frac{1}{2}} ww^\top \widehat{\Sigma}_y^{\frac{1}{2}} / \gamma) = (1 - w^\top \widehat{\Sigma}_y w / \gamma).$$

Notice that the constraint $\gamma \widehat{\Sigma}_y^{-1} \succ ww^\top$ is translated into $\gamma > w^\top \widehat{\Sigma}_y w$. Thus, we have

$$\tau_y^{\mathrm{D}}(w)^2 = \inf_{\gamma > w^\top \widehat{\Sigma}_y w} \gamma \rho_y - \gamma \log(1 - w^\top \widehat{\Sigma}_y w / \gamma).$$

The first-order optimality condition for $\gamma$ is

$$\rho - \log\left(1 - \frac{w^\top \widehat{\Sigma}_y w}{\gamma}\right) - \frac{w^\top \widehat{\Sigma}_y w}{\gamma - w^\top \widehat{\Sigma}_y w} = 0,$$

and the optimal solution for $\gamma$ is

$$\gamma^\star = \frac{w^\top \widehat{\Sigma}_y w}{1 + 1/W_{-1}(-\exp(-\rho_y - 1))}.$$

Replacing the value of $\gamma^\star$ into the objective function leads to the necessary result. □

