# OpenReview forum: "Covariance-Robust Minimax Probability Machines for Algorithmic Recourse"
_ICLR.cc/2023/Conference — Submitted to ICLR 2023_

### Official Review · Reviewer_14wF · 2022-10-21

**Confidence:** 4
**Correctness:** 2
**Technical Novelty And Significance:** 3
**Empirical Novelty And Significance:** 2
**Recommendation:** 3

**Clarity, Quality, Novelty And Reproducibility:**

I found a few things unclear:
- "By definition $\mathbb{Q}$ ... equals zero if and only if $\Sigma = \hat \Sigma$. I don't think this is by definition; this follows by its relation to the Frobenius norm.
- In the definition of $\mathbb{Q}$, is the square meant to indicate entrywise squaring or a matrix product?
- I can't tell what the $\sim$ notation means. It looks like a distribution always goes on the left hand side, but sometimes a set of parameters go on the right, and sometimes a distribution goes on the right.
- In Section 6.1, a max is taken over $x' \sim N(x, 0.001I)$. Is this a max over the support over the normal distribution (so a max over all real numbers)?
- In the definition of LocalFid, I don't think $s_x$ was ever defined; this should be explicitly defined.


**Strength And Weaknesses:**

The provided experiments are interesting, and provide good evidence that their proposed method can outperform existing methods. I overall feel that the proposed method needs some more investigation and explanation before publication. I've tried to break issues down into categories below:


**Theoretical development.** A number of theorems and propositions are stated about fitting various MPMs and the properties of MPMs. I found three proofs that were explicitly omitted and a couple more that I had a hard time following. First, I do not think it is OK to publish a paper with omitted proofs. Omitting proofs makes the paper extremely difficult to validate, reproduce, and build on in future work; I definitely cannot tell if the stated results are correct or not. I read through the proof of Proposition 3.4, and got stuck in a few places:
1. "note that the worst-case probability admits a two-layer decomposition" I don't immediately see why this is true. I don't think the notation of taking a max over $P_y \sim (\hat\mu_y, \Sigma_y)$ was ever defined.
2. The proof ends without reducing to something that looks like the objective $\min_{w \in W} \sum_y \tau_y^\varphi (w)$. Is there some kind of extra dual argument needed? All of the steps should be made clear in the proof.

I couldn't follow the proof of Proposition 3.5 for the same reasons as Proposition 3.5; I didn't check the rest of the proofs.

**Motivation.** Some of the results didn't feel well-motivated.
1. It seems that one major addition over Lanckriet et al. (2003) is the use of new metrics over covariance matrices in the MPM. But it's not clear why this is desirable. The authors note "the [previously used] quadratic divergence ... does not coincide with any distance between probability distributions". Why is this something we should want given the goal of producing algorithmic recourses that are stable to model shift? Why study both the Bures and Fisher-Rao MPM?
2. What is the purpose of the asymptotics in the paper? It seems like the asymptotic regime studied requires at least one of the radii of the ambiguity sets to grow to infinity. Why should we be concerned about the model's performance when we are infinitely uncertain about things? What do the derived limits tell us about the suitability of these MPMs for algorithmic recourses?
3. I'm not exactly sure what the experiments tell us about the proposed method. In Section 6.2, in some cases, it looks like one of the proposed methods is outperforming competing methods according to some metrics. Why is this happening? What about the datasets is causing FR-MPM-PROJ to succeed? Additionally, the experiments in Figure 4 only show the use of the Fisher-Rao distance; what about using Bures or quadratic distances?

**Summary Of The Paper:**

This paper studies the problem of providing algorithmic recourses -- that is, providing users who were rejected by a model (e.g. denied a loan) a way to change themselves so that they will not be rejected in the future. The paper is specifically concerned with the setting where the model may change over time (e.g. by getting updated training data); in this setting, we would want old recourses to be valid for the new model. The authors propose to locally fit a linear surrogate model around the given model's decision boundaries. They specifically use "minimax probability machines" (MPMs) to make these linear surrogate models more robust to changing models. The authors propose a few modifications to existing MPMs; in particular, they propose different objective functions in fitting the MPMs and study the asymptotic behaviors of the resulting surrogate models. With a linear surrogate in hand, existing algorithmic recourse methods are applied to the surrogate model.

**Summary Of The Review:**

I mainly think the paper needs to contain proofs for all of its claimed results and explain what the takeaways are from all of its claimed results. I think both of these things need to be included in a paper, so I've voted for reject.

---

> ### Author Response · Authors · 2022-11-19
> **Response to Reviewer 14wF [Part 1]**
>
> We thank the reviewer for taking the time to evaluate our work and provide great constructive feedback. We tried our best to revise the paper according to the reviewer’s suggestions for clarity. We address the remaining concerns of the reviewer as follows.
>
> ### Theoretical development
>
> We have revised the all proofs and made changes for clarity, particularly for the proof of Proposition 3.4 and 3.5. We expanded the explanations for the two-layer decompositions and included the derivations to the proposition’s claims. We also included all omitted proofs in the paper and it is self-contained now.
>
> ### Empirical contributions
>
> > **TL;DR** We would also like to emphasize again that our paper has significant empirical contributions to the robust recourse literature.
> A comprehensive evaluation of common linear surrogates (LIME, CLIME, and LIMELS) on typical recourse generators (ROAR and AR).
> Propose to impose robustness to the surrogate and use a simple recourse search to generate *actionable* and *robust* recourses.
>
> It’s worth noting that the local linear surrogate is commonly used in recourse literature [3, 4, 5]. However, the impact of the linear surrogate on recourse generation is still unknown. This paper made the first attempt to study it by providing a comprehensive evaluation of the common linear surrogates (including LIME, CLIME which is designed to improve the stability of the surrogates, and LIMELS which is designed to improve the fidelity of the surrogates) to typical research generators such as ROAR and AR.
>
> Moreover, the conventional approach to generating robust recourses is to first approximate a non-robust surrogate and then construct robust recourses with the surrogate model. We propose to switch the viewpoint by injecting robustness into the linear surrogate instead of the recourse generation step. That facilitates the generation of robust recourses while still being able to incorporate AR to promote actionability, which is hard for gradient-based approaches like ROAR. Extensive experimental results presented in our paper suggest that FR-MPM could provide stable and faithful surrogates while consistently providing a better balance in the trade-off between cost and validity (robustness) of generated recourses.
>
> Another significant aspect is that our method could incorporate AR to promote actionability which is hard for gradient-based approaches like ROAR. Note that actionability is crucial to employ recourses in real-world applications. Table 2 and Table 5 show that our method could improve both current validity and future validity substantially compared to AR using LIME-based surrogates.
>
> ### Motivation
>
> **Q1: Why connection to distances between probability distributions is needed? Why study both the Bures and Fisher-Rao MPM?**
>
> Many studies have shown that real-world data often lie on geometric shapes (e.g., low-dimensional manifolds). When data perturbations are quantified, one should respect the data geometry. In our model, the samples are used via their covariance matrix. The adoption of a non-Euclidean distance for covariance matrices can therefore be seen as an acknowledgment and exploitation of data geometry.
>
> Moreover, the studies of Bures and Fisher-Rao MPMs provide new tools for estimating the surrogate with different properties. We added Section A.3 in the appendix to compare covariance-robust MPMs with different distances and provide guidance for choosing the surrogate in practice. We will elaborate on this when answering your next question using the results of asymptotic MPMs.

---

> > ### Author Response · Authors · 2022-11-19
> > **Response to Reviewer 14wF [Part 2]**
> >
> > **Q2: What is the purpose of the asymptotics in the paper? What do the derived limits tell us about the suitability of these MPMs for algorithmic recourses?**
> >
> > > **TL;DR** The purpose of the asymptotic MPMs is to study the behaviors of covariance MPMs with different distances.
> > Suitability of these MPMs for algorithmic recourses: when querying the underlying model is not expensive, so we can sample sufficient data to estimate the covariance matrices accurately, FR-MPM would be preferred. Otherwise, we should use Quadratic or Bures MPMs.
> >
> > Proposition 4.4 and Remark 4.5 showed that Quadratic MPM and Bures MPM coincide when one of the radii $\rho_y$ grows to infinity, and they are independent of the covariance matrices $\hat\Sigma_y$. Meanwhile, the asymptotic hyperplane of the Fisher-Rao MPM when $\rho_y \to \infty $ aligns with axes of the covariance matrices $\hat\Sigma_y$ (see Proposition 5.4 and Figure 14). It suggests that the Fisher-Rao MPM is not a suitable surrogate at low sample sizes as it relies on the estimate of the covariance matrices. On the other hand, when the number of samples is sufficient to estimate the covariance matrices accurately, Fisher-Rao MPM would be better than Quadratic MPM and Bures MPM as it takes the geometry of the data into account when robustifying the surrogate.
> >
> > We also provided experimental results in Appendix A.3 to probe the fidelity and robustness of constructed recourses upon MPM surrogates at low sample sizes. The results showed that FR-MPM shows similar local fidelity with Quadratic MPM and Bures MPM when the covariance matrices are ill-conditioned. When increasing the number of samples, Fisher-Rao MPM significantly outperforms Quadratic MPM and Bures MPM. The recourses constructed with Fisher-Rao MPM also exhibit a better cost-validity trade-off.
> >
> >
> > **Q3: Why does FR-MPM-PROJ outperform other baselines? What about the datasets that are causing FR-MPM-PROJ to succeed?**
> >
> > > **TL;DR** Wachter might suffer from local optimal, and it does not consider the robustness of recourses, while ROAR might be overly conservative due to the minimax robust optimization objective. Our methods address these drawbacks by solving distributionally robust MPMs and using a simple and efficient projection to generate recourses.
> >
> > * In this paper, we mainly compare our methods (MPM-related methods) to Wachter and ROAR with different non-robust surrogate models.
> > Wachter is the method to construct recourses for non-linear models but requires access to the model's gradient. It minimizes a weighted-sum objective between the recourse's feasibility and cost, which might be non-convex when the classifier is a non-linear model. Furthermore, Wachter does not consider the underlying model's uncertainty. We conduct a careful evaluation for Wachter by varying the hyperparameter $\lambda$ (Figure 9, Appendix A.2) and the probabilistic threshold that is used to define a favorable region (often set to 0.5) (Figure 12, Appendix A.2). The results show that our MPM-related methods consistently outperform Wachter for all evaluated datasets.
> > * It is known that a robust optimization framework like ROAR could be overly conservative because it may hedge against a pathological parameter in the uncertainty set [5]. Our methods instead solve distributionally robust MPMs hedging changes in the covariance matrices, thus taking data distribution into account when imposing robustness. Thereby, we use a simple and cost-efficient projection to construct robust recourses. Furthermore, our covariance-robust MPMs are also shown to have higher fidelity than LIME-related surrogates, which is crucial for generating recourses.
> >
> > We compare our methods with ROAR on three real-world datasets used in their paper, similar to [3, 4]: German credits, SBA, and Student. These datasets are commonly used in robust recourse literature as they can capture natural shifts in real-world scenarios such as the correction shift (German credit), the temporal shift (SBA), or the geospatial shift (Student).
> >
> > **Q4: Additionally, the experiments in Figure 4 only show the use of the Fisher-Rao distance; what about using Bures or quadratic distances?**
> >
> > In the main paper, we focus on illustrating the performance of the Fisher-Rao method compared to existing methods (LIME, etc.). This choice is solely for aesthetic purposes: Figure 4 clearly depicts the superior performance of the FR method.
> >
> > The performance of the Bures and Quadratic MPM are provided in Appendix A.2, Figure 13.

---

> > > ### Author Response · Authors · 2022-11-19
> > > **Response to Reviewer 14wF [Part 3]**
> > >
> > > ### Clarity
> > >
> > > **Q1: By definition Q ... equals zero if and only if Σ=Σ^...**
> > >
> > > We revised this paragraph according to the reviewer's suggestion.
> > >
> > > **Q2: In the definition of Q, is the square meant to indicate entrywise squaring or a matrix product?**
> > >
> > > Square here means the matrix product. Similarly, in the definitions of the Bures and Fisher-Rao divergences, the square root of $\Sigma$ indicates the matrix square root.
> > >
> > > **Q3: Notation ~**
> > >
> > > We apologize for the confusion. We have removed the $x^\prime \sim \mathcal(N, 0.0001I)$ in Section 6.1. Now, we only use $\mathbb{P} \sim (\mu, \Sigma)$ to denote that the distribution $\mathbb{P}$ has mean vector $\mu$ and covariance matrix $\Sigma$.
> > >
> > > **Q4: In Section 6.1, max is taken over x′∼N(x,0.001I)...**
> > >
> > > We have revised this section for clarity. We use the Stability metric, similar to [Agarwal et al. 2021] to measure the stability of the surrogate with respect to small perturbations in the input instance. For a given instance $x$, we draw a set $\mathcal{U}\_x$ of neighbors of $x$ independently from a Gaussian distribution $\mathcal{N}(0, 1)$. We generate a surrogate $\theta_{x^\prime}$ for each neighbor $x^\prime \in \mathcal{U}$ and report the maximum distance between the explanation of $x$ to that of $x’$.
> > >
> > > The Local Fidelity metric [Laugel et al., 2018] measures the agreement of the surrogate to the underlying classifier. We draw 1000 samples from the unit ball around a given input instance and report the fraction of samples where the output of the underlying classifier and the surrogate agree.
> > >
> > > **Q5: Definition of s_x in the LocalFid metric**
> > >
> > > We thank the reviewer. We fixed the notation of the linear surrogate to $\mathcal{C}_{\theta} (x) = \mathrm{sign}(w^T x - b)$, where $\theta = (w, b)$ to synchronize to Section 2.
> > >
> > > We hope that we have addressed all your concerns adequately. If you have any further remaining concerns, please do not hesitate to ask. We are looking for a positive response from the reviewer.
> > >
> > > **References**
> > >
> > > [1] Ben-Tal, Aharon, et al. "Globalized robust optimization for nonlinear uncertain inequalities." INFORMS Journal on Computing 29.2 (2017): 350-366.
> > >
> > > [2] Pawelczyk, Martin, et al. "Carla: a python library to benchmark algorithmic recourse and counterfactual explanation algorithms." arXiv preprint arXiv:2108.00783 (2021).
> > >
> > > [3] Upadhyay, Sohini, Shalmali Joshi, and Himabindu Lakkaraju. "Towards robust and reliable algorithmic recourse." Advances in Neural Information Processing Systems 34 (2021): 16926-16937.
> > >
> > > [4] Bui, Ngoc, Duy Nguyen, and Viet Anh Nguyen. "Counterfactual Plans under Distributional Ambiguity." arXiv preprint arXiv:2201.12487 (2022).
> > >
> > > [5] Ustun, Berk, Alexander Spangher, and Yang Liu. "Actionable recourse in linear classification." Proceedings of the conference on fairness, accountability, and transparency. 2019.

---

> > > > ### Author Response · Authors · 2022-12-09
> > > > **For any further questions!**
> > > >
> > > > Dear reviewer,
> > > >
> > > > We thank the reviewer again for your insightful reviews and thoughtful suggestions. We have done our best to address the questions raised in your review and hope to have a further discussion with you to see if our responses adequately address your concerns. The discussion period is coming to a close in the next few days and we remain open to discussing any remaining concerns you may have until the very end. If there are any remaining questions or concerns, please do not hesitate to ask.
> > > >
> > > > Thank you for taking the time to evaluate our work and our responses.
> > > >
> > > > Authors

---

### Official Review · Reviewer_33C8 · 2022-10-24

**Confidence:** 3
**Correctness:** 4
**Technical Novelty And Significance:** 3
**Empirical Novelty And Significance:** 3
**Recommendation:** 8

**Clarity, Quality, Novelty And Reproducibility:**

The paper's writing is clear and easy to understand - please see my comments above.

The paper's quality is good, and I think the proposed approach is novel and can be of benefit to the community. Theoretical analyses are also conducted to understand the performance of the proposed approach (although note I can't dig into detail to confirm the correctness of these analyses).

The code is made available so I believe in the reproducibilty of the paper.

**Strength And Weaknesses:**

Strengths:
+ The paper targets an important and interesting problem. The related work and existing literature are described in detailed and thorough.
+ The paper's writing is very clear and easy to understand. Even though the paper has many complicated theoretical analyses, but thanks to the clear writing, these analyses could be followed easily.
+ A lot of theoretical analyses are derived to understand the property of the MPM-based proposed approach. They seem to be sound to me, although note that I can't dig into detail of the proof, so I can't guarantee if these theoretical analyses are completely correct or not.
+ The discussion around the experimental evaluation is thorough and detailed. The paper also compared their proposed approach with various existing and new baselines. Different metrics are used to evaluate different aspects of the proposed approach.
+ The code is published in order to aid reproducibility.

Weaknesses:
+ The datasets used in the experimental evaluation seem to be simple and easy datasets. Although I completely understand that the research field of algorithmic recourse is new, and thus the evaluation can be done on some simple datasets.

Questions:
+ Based on the experimental evaluation, it seems to me that the approach of using a robust surrogate then derive a recourse from this robust surrogate is much better compared to the approach of using a non-robust surrogate and then derive a robust recourse. Is there any explanation for this particular property? Does this mean the choice of surrogate model affects robustness property the most?
+ There are several settings in the experiments that I feel unclear on why they are set that way. For example, in Section 6.1 - Stability, 10 neighbours data points are sampled in the distribution N(x, 0.001I). In Section 6.1 – Fidelity, r_fid is set to 10% of the maximum distance between instances in the training data. Or the radius r_p is set to 5% of the maximum distance between instances in the training data. Why are these values (10, 0.001, 10%, 5%) chosen? Will these values be sensitive to the performance of the proposed approach?


**Summary Of The Paper:**

The paper targets the problem of algorithmic recourse, which is to suggest how an input instance should be modified to alter the outcome of a predictive model. In particular, the paper proposes a pipeline to generate a model-agnostic recourse that is robust to model shifts. The main idea is to estimate a linear surrogate of the black-box model using covariance-robust minimax probability machines (MPM), then generate the recourse using this surrogate. The paper derives several theoretical analyses to show that the covariance-robust MPM correspond to the l2-regularization and class-reweighting schemes. Finally, experiments are conducted on various real-world datasets to confirm the effectiveness of the proposed approach.

**Summary Of The Review:**

Overal, I think the paper targets an important problem. The key idea seems to be sound and novel to me. The experiments are also detailed and thorough. The paper's writing is very clear, and easy to understand.

---

> ### Author Response · Authors · 2022-11-19
> **Response to Reviewer 33C8**
>
> Thank you for your thoughtful review and valuable feedback. Below we address your concerns.
>
> **Q1: The chosen datasets**
>
> We use three datasets: German credits, SBA, and Student, similar to [3, 4]. These datasets are commonly used in robust recourse literature as they can capture natural shifts in real-world scenarios such as the correction shift (German credit), the temporal shift (SBA), or the geospatial shift (Student).
>
> **Q2: Why does injecting robustness to the surrogate have better results than injecting to the recourse generation phase? Does this mean the choice of surrogate model affects robustness property the most?**
>
> Injecting robustness to the surrogate by covariance-robust MPMs has several advantages compared to ROAR:
>
> * We provide robust surrogates with high stability and fidelity that facilitate the generation of robust recourses. Being faithfulness to the decision boundary of the underlying model is crucial to generate recourses with a better cost-validity trade-off.
> * It is known that a robust optimization framework like ROAR could be overly conservative because it may hedge against a pathological parameter in the uncertainty set [5]. Our methods instead solve distributionally robust MPMs hedging changes in the covariance matrices, thus taking data distribution into account when imposing robustness. Thereby, we just use a simple and cost-efficient projection to construct robust recourses.
> * Another significant aspect is that our method could incorporate AR to promote actionability which is hard for gradient-based approaches like ROAR. Note that actionability is crucial to employ recourses in real-world applications.
>
> We would like to emphasize again that our experiments focus on the trade-off between cost and robustness of recourses, not just robustness only as one could easily produce a costly robust recourse. Therefore, we would not claim that the choice of the surrogate affects the robustness of the constructed recourse the most. Instead, we showed that by switching the viewpoint of robustness, we could construct a more stable and faithful surrogate that facilitates a robust generation of actionable recourses with a better cost-validity trade-off.
>
> **Q3: The choice of experimental settings. Will these values be sensitive to the performance of the proposed approach?**
>
> In Section 6.1, we choose the settings for the stability metric according to [1] and for local fidelity according to [2]. We also rerun the experiment in Section 6.1 with a different setting and present the result in Figure 8, Appendix A.2. Specifically, $10$ neighbors are sampled in the distribution $\mathcal{N}(x, 0.0001I)$ to measure the stability. Meanwhile, $r_{fid}$ is set to $20$\%, and the radius $r_p$ is set to $10$\% of the maximum distance between instances in the training data. The result is pretty similar to the previous setting.
>
> We hope that we have cleared your concerns about our paper. If you have any further remaining questions, please do not hesitate to ask. We are looking for further feedback from the reviewer.
>
> **References**
>
> [1] Agarwal, Sushant, et al. "Towards the unification and robustness of perturbation and gradient based explanations." International Conference on Machine Learning. PMLR, 2021.
>
> [2] Laugel, Thibault, et al. "Defining locality for surrogates in post-hoc interpretablity." arXiv preprint arXiv:1806.07498 (2018).
>
> [3] Upadhyay, Sohini, Shalmali Joshi, and Himabindu Lakkaraju. "Towards robust and reliable algorithmic recourse." Advances in Neural Information Processing Systems 34 (2021): 16926-16937.
>
> [4] Bui, Ngoc, Duy Nguyen, and Viet Anh Nguyen. "Counterfactual Plans under Distributional Ambiguity." arXiv preprint arXiv:2201.12487 (2022).
>
> [5] Ben-Tal, Aharon, et al. "Globalized robust optimization for nonlinear uncertain inequalities." INFORMS Journal on Computing 29.2 (2017): 350-366.

---

> > ### Author Response · Authors · 2022-12-09
> > **For any further questions!**
> >
> > Dear reviewer,
> >
> > We thank the reviewer again for your insightful reviews and thoughtful suggestions. We have done our best to address the questions raised in your review and hope to have a further discussion with you to see if our responses adequately address your concerns. The discussion period is coming to a close in the next few days and we remain open to discussing any remaining concerns you may have until the very end. If there are any remaining questions or concerns, please do not hesitate to ask.
> >
> > Thank you for taking the time to evaluate our work and our responses.
> >
> > Authors

---

### Official Review · Reviewer_NHFj · 2022-10-24

**Confidence:** 4
**Correctness:** 3
**Technical Novelty And Significance:** 2
**Empirical Novelty And Significance:** 1
**Recommendation:** 3

**Clarity, Quality, Novelty And Reproducibility:**

The paper is overall well-written, however, not so novel. The authors provide the code, but I haven’t checked it.

**Strength And Weaknesses:**

### Strength:
* The MPM part is easy to follow.

### Weaknesses:
* From my point of view, I don’t think the derivation of the covariance-robust MPMs with these two statistical divergences is hard. And I think an extension to general f-divergence is also possible.
* The authors spend most of the space on presenting the variants of covariance-robust MPMs, without discussing the necessity of studying these variants for algorithmic recourse.
* The empirical results do not have significant improvement over the quadratic MPM.


**Summary Of The Paper:**

This paper introduces two variants of covariance-robust minimax probability machines (MPM) which the authors term as Bures MPM and Cramer-Rao MPM based on different statistical divergences between Gaussian distributions. The authors discuss how to estimate the parameter for these new variants of MPM and discuss the potential applications for algorithmic recourse.

**Summary Of The Review:**

The authors do not provide strong motivation for proposing the two new MPM variants. The theoretical results are not technically hard and the empirical results are not strong.

---

> ### Author Response · Authors · 2022-11-19
> **Response to Reviewer NHFj [Part 1]**
>
> Thank you for reviewing and providing feedback on our paper. We address your concerns as the following:
>
> **Q1: Novelty.**
>
> **Empirical novelty.**
>
> We would also like to emphasize again that our paper has significant empirical contributions to the robust recourse literature.
>
> It’s worth noting that the local linear surrogate is commonly used in recourse literature [1, 2, 3]. However, the impact of the linear surrogate on recourse generation is still unknown. This paper made the first attempt to study it by providing a comprehensive evaluation of the common linear surrogates (including LIME, CLIME which is designed to improve the stability of the surrogates, and LIMELS which is designed to improve the fidelity of the surrogates) to typical research generators such as ROAR and AR.
>
> Moreover, the conventional approach to generating robust recourses is to first approximate a non-robust surrogate and then construct robust recourses with the surrogate model. We propose to switch the viewpoint by injecting robustness into the linear surrogate instead of the recourse generation step. That facilitates the generation of robust recourses while still being able to incorporate AR to promote actionability, which is hard for gradient-based approaches like ROAR. Extensive experiments in our paper suggest that FR-MPM could provide stable and faithful surrogates while consistently providing a better balance in the trade-off between cost and validity (robustness) of generated recourses.
>
> Another significant aspect is that our method could incorporate AR to promote actionability which is hard for gradient-based approaches like ROAR. Note that actionability is crucial to employ recourses in real-world applications. The results in Table 2 and Table 5 show that our method could improve both current validity and future validity substantially compared to AR using LIME-based surrogates.
>
> **Theoretical novelty.**
>
> ​​While most contributions in this field have been established by the seminal work of Lanckriet et al in multiple papers, it remains open how we can incorporate regularization into the MPM framework. Our paper establishes this connection and we recover the norm regularization and the reweighting scheme through a principled approach of robustifying against the covariance matrix perturbations. We would like to highlight that establishing the connection between regularization and the distributionally robust optimization (DRO) problem can lead to new insights and is of particular interest in machine learning (especially in robust/safe/adversarial ML). Previously, for the empirical risk minimization problem, we understood that optimal transport DRO can recover norm regularization [4], while f-divergence DRO leads to reweighting [5]. This paper extends these connections to the MPM problem.

---

> > ### Author Response · Authors · 2022-11-19
> > **Response to Reviewer NHFj [Part 2]**
> >
> > **Q2: Extension to f-divergence.**
> >
> > This is an excellent observation. We agree with the reviewer that our approach can be extended to a much bigger class of divergence. However, in order to have an analytical expression for $\tau^{\varphi}_y (w)$ like in the Bures and Fisher-Rao cases, we need to exploit the structural assumptions on the divergence $\varphi$. Our derivation also depends critically on the availability of the explicit formular for the divergence. In our opinion, ageneralization is out of the scope of the paper and requires substantial extra details, we will leave it to future publications.
> >
> > **Q3: Empirical results do not have significant improvement over Quadratic MPM.**
> >
> > We emphasize that the goal of this paper is to compare (covariance-robust) MPM versus existing methods in the field of robust recourse. We highlight once again our contributions in this order:
> >
> > 1. The use of MPM in algorithmic recourse. Note that no prior work in this field has mentioned the possibility of MPM for constructing a linear surrogate.
> >
> > 2. The intuitive shift of the surrogate, and its application in robust recourse. This shift is induced by a parameter $\rho$. Note that no prior work even in the field of MPM has mentioned the implications of this shift, but in our paper, we provide an interesting use case in which this shift can lead to robust recourses.
> >
> > 3. The connection of robust MPM to regularization. We bring up the Bures and Fisher-Rao models here simply to illustrate that robustness is equivalent to regularization, which is an interesting research topic for the field of robust optimization.
> >
> > Thus, showing that Bures or Fisher-Rao MPM can dominate Quadratic MPM is not the ultimate goal of this paper. Instead, we should view these three methods as complementary approaches that dominate the current practice of using LIME or LIMELS as a linear surrogate.
> >
> > To compare MPM with different distances, we added Section A.3 in the appendix to discuss the variants of covariance-robust MPMs and provide guidance for choosing the surrogate model in practice, especially at a low sample size.
> >
> > We hope that our responses have addressed your concerns. We are happy to answer in case you have any further questions.
> >
> > **References:**
> >
> > [1] Upadhyay, Sohini, Shalmali Joshi, and Himabindu Lakkaraju. "Towards robust and reliable algorithmic recourse." Advances in Neural Information Processing Systems 34 (2021): 16926-16937.
> >
> > [2] Bui, Ngoc, Duy Nguyen, and Viet Anh Nguyen. "Counterfactual Plans under Distributional Ambiguity." arXiv preprint arXiv:2201.12487 (2022).
> >
> > [3] Ustun, Berk, Alexander Spangher, and Yang Liu. "Actionable recourse in linear classification." Proceedings of the conference on fairness, accountability, and transparency. 2019.
> >
> > [4] Shafieezadeh-Abadeh, Soroosh, Daniel Kuhn, and Peyman Mohajerin Esfahani. "Regularization via mass transportation." Journal of Machine Learning Research 20.103 (2019): 1-68.
> >
> > [5] Duchi, John, and Hongseok Namkoong. "Variance-based regularization with convex objectives." The Journal of Machine Learning Research 20.1 (2019): 2450-2504.

---

> > > ### Author Response · Authors · 2022-12-09
> > > **For any further questions!**
> > >
> > > Dear reviewer,
> > >
> > > We thank the reviewer again for your insightful reviews and thoughtful suggestions. We have done our best to address the questions raised in your review and hope to have a further discussion with you to see if our responses adequately solve your concerns. The discussion period is coming to a close in the next few days and we remain open to discussing any remaining concerns you may have until the very end. If there are any remaining questions or concerns, please do not hesitate to ask.
> > >
> > > Thank you for taking the time to evaluate our work and our responses.
> > >
> > > Authors

---

### Official Review · Reviewer_CMDf · 2022-10-29

**Confidence:** 2
**Correctness:** 4
**Technical Novelty And Significance:** 3
**Empirical Novelty And Significance:** 3
**Recommendation:** 8

**Clarity, Quality, Novelty And Reproducibility:**

## Clarity
- A schematic of the sampling done in step 1 of the algorithm would improve the readability of the algorithm description. Figure 2 partially, but not fully, addresses this.
- It feels to me there is a big jump from the introduction related to algorithmic recourse to the development of the method. I think a sentence or two better motivating the choices made in the method would improve the exposition, e.g. how robustness of the MPM translates back to robustness of a recourse, would be helpful.

## Quality and novelty
- I am not familiar enough with related work to comment on the novelty of the ideas in relation to existing literature. The quality of the experiments and derivations seems to be reasonably high.

## Reproducibility
- Experimental details are provided, and existing code relied on for experiments is documented, so it seems likely the findings are reproducible. Code is provided, as well as details to run the code.

**Strength And Weaknesses:**

## Strengths
- The experiments appear to be quite thorough, especially considering the detail provided in the supplement.
- The method seems to be a reasonable minor adaptation of existing methods in MPM, and yet some benefit is still achieved in experiments. This suggests it might be quite practical.
- The presentation is generally good.

## Weaknesses

- The authors claim that a major contribution of the work is an "intuitive and interpretable approach to generate robust recourse". However, not much of the paper is dedicated to the interpretation of what it means for robustness to be defined with respect to a MPM that is distributionally robust with respect to a particular distance between Gaussians. For example, how different are the Bures and Quadratic distances in practice; would figure 2 change much had you used quadratic distance? Intuitively when would I want to use a Bures MPM versus a Fisher-Rao MPM? In the fidelity/stability experiments, it seems the Fisher-Rao MPM often is more stable/higher fidelity. Can you convey intuition as to why this is the case? I think the geometric intuition in figure 2 and remark 4.6 is helpful, but you should go further in relating the distances chosen to robustness of the resulting recourse if a major claim of the paper is that the method is both intuitive and interpretable.


**Summary Of The Paper:**

The authors consider the application of a distributionally robust minimax probability machine, relying on statistical distances defined on the space of Gaussian measures. The authors motivate this as a tool for finding algorithmic recourses that are robust to changes in parameters of a classifier, that might be induced by underlying changes in the distribution of covariates. Several experiments are considered to compare the method in terms of a cost-validity trade-off and actionability of the recourses.

**Summary Of The Review:**

I am in favor of accepting the paper on balance. I think the presentation was reasonably good, the method seems practical and experiments seem thorough. However, I am not an expert in this area and have not read large portions of the related literature. I therefore have a low degree of confidence in my review and it is very possible I will change my score in light of future feedback and discussion with the other reviewers and authors.

---

> ### Author Response · Authors · 2022-11-19
> **Response to Reviewer CMDf**
>
> We thank the reviewer for taking the time to evaluate our paper. We answer your questions as follows:
>
> **Q1: Interpretation of covariance-robust variants of MPM w.r.t. different distances.**
>
> Many studies have shown that real-world data often lie on geometric shapes (e.g., low-dimensional manifolds). When data perturbations are quantified, one should respect the data geometry. In our model, the samples are used via their covariance matrix. The adoption of a non-Euclidean distance for covariance matrices can therefore be seen as an acknowledgment and exploitation of data geometry.
>
> **Q2: Would figure 2 change much had you used quadratic distance?**
>
> We have provided another illustration to compare asymptotic hyperplanes of covariance-robust MPM with different distances in Figure 14, Appendix A2. Remark 4.5 suggests that the hyperplane of Quadratic MPM comes close to the hyperplane of Bures MPM when the uncertainty radii grow to infinity. Meanwhile, the hyperplane of Fisher-Rao MPM would be different as its asymptotic hyperplane takes the covariance matrices into account (Proposition 5.4). It is consistent with the observation in Figure 14.
>
> **Q3: When would I want to use a Bures MPM versus a Fisher-Rao MPM?**
>
> We thank the reviewer for this intriguing question. We added Section A.3 in the appendix to compare covariance-robust MPMs with different distances and provide guidance for choosing the surrogate in practice in Appendix A.3.
>
> The arguments rely on the comparison of asymptotic MPM’s hyperplanes. Proposition 4.4 and Remark 4.5 showed that Quadratic MPM and Bures MPM coincide when one of the radii $\rho_y$ grows to infinity and they are independent of the covariance matrices $\hat \Sigma_y$. Meanwhile, the asymptotic hyperplane of the Fisher-Rao MPM when $\rho_y \to \infty $ aligns with axes of the covariance matrices $\hat \Sigma_y$ (see Proposition 5.4 and Figure 14). It suggests that the Fisher-Rao MPM is not a suitable surrogate at low sample sizes as it relies on the estimate of the covariance matrices. On the other hand, when the number of samples is sufficient to estimate the covariance matrices accurately, Fisher-Rao MPM would be better than Quadratic MPM and Bures MPM as it takes the geometry of the data into account when robustifying the surrogate.
>
> We also provided experimental results in Appendix A.3 to probe the fidelity and robustness of constructed recourses upon MPM surrogates at low sample sizes. The results showed that FR-MPM shows similar local fidelity with Quadratic MPM and Bures MPM when the covariance matrices are ill-conditioned. When increasing the number of samples, Fisher-Rao MPM outperforms Quadratic MPM and Bures MPM significantly. The recourses constructed with Fisher-Rao MPM also exhibit a better cost-validity trade-off.
>
> **Q4: Why does FR-MPM have better stability and fidelity compared to QUAD-MPM and BW-MPM?**
>
> As mentioned above, Propositions 4.4 and 4.5 suggest that asymptotic hyperplanes of Quadratic and Bures MPM are independent of the covariance matrices while that of Fisher-Rao does take the covariance matrix into account. It could be seen as a better exploitation of data geometry when robustifying the surrogate.
>
> **Q5: Connection between chosen distances to resulting recourses.**
>
> In our framework, the recourse generation depends on two components: the surrogate linear model and the recourse search (projection onto the hyperplane). In Appendix A.3 (added in the revision), we try to provide some information on how the surrogate depends on the distances (Quad, Bures, and Fisher-Rao). In Figure 14, we also depict a qualitative difference between the different hyperplanes as the radius changes.
>
>
>
> **Clarity**
>
> **Q1: A schematic of the sampling in step 1.**
>
> We thank the reviewer for the constructive suggestion. We updated Figure 1 to aid the illustration of our algorithm.
>
> **Q2: …I think a sentence or two better motivating the choices made in the method would improve the exposition.**
>
> We thank the reviewer for the suggestion. We have included an intuition explanation of the robustification mechanism in Figure 2. We also generate an animated GIF in our repo (and supplementary material) to aid the visualization.
>
> We thank the reviewer again for their thoughtful comments and feedback. We hope we have addressed all your questions/concerns/comments adequately. We are glad to answer any further questions that you have about our paper.

---

> > ### Author Response · Authors · 2022-12-09
> > **For any further questions!**
> >
> > Dear reviewer,
> >
> > We thank the reviewer again for your insightful reviews and thoughtful suggestions. We have done our best to address the questions raised in your review and hope to have a further discussion with you to see if our responses adequately address your concerns. The discussion period is coming to a close in the next few days and we remain open to discussing any remaining concerns you may have until the very end. If there are any remaining questions or concerns, please do not hesitate to ask.
> >
> > Thank you for taking the time to evaluate our work and our responses.
> >
> > Authors

---

### Author Response · Authors · 2022-11-19
**General Response**

Dear AC and reviewers,

We appreciate your efforts in reviewing and providing many constructive feedbacks to our paper!

Several reviewers express a common concern about the distinction among Quadratic, Bures and Fisher-Rao MPM, and ask for a rigorous/theoretical justification for a performance ranking among these three methods. This concern is valid, however, it should not be a drawback to overlook the main contributions of our paper.

We highlight our contributions in this order:

1. The use of MPM in algorithmic recourse. Note that no prior work in this field has mentioned the possibility of MPM for constructing a linear surrogate.

2. The intuitive shift of the surrogate, and its application in robust recourse. This shift is induced by a parameter $\rho$. Note that no prior work even in the field of MPM has mentioned the implications of this shift, but in our paper, we provide an interesting use case in which this shift can lead to robust recourses.

3. The connection of robust MPM to regularization. We bring up the Bures and Fisher-Rao models here simply to illustrate that robustness is equivalent to regularization, which is an interesting research topic for the field of robust optimization.

In the revised manuscript, we try to shed some lights on the comparison between the three methods (see Appendix A3). We also note that the scientific community has collectively spent a tremendous effort in order to study the difference between regularization schemes ($l_1$, $l_2$, reweighting, etc.), and thus we believe that a definitive answer to this issue (which type of robust MPM is the best?) is beyond the scope of this paper. We note that our contribution 3 above can form new bridges to answer this question (by merging knowledge from robustness and regularization).

Thank you very much for your consideration!

---

### Author Response · Authors · 2022-12-01
**Thanks all reviewers!**

We would like to thank all reviewers for your insightful reviews and suggestions. We have revised our manuscript and added new responses to your comments and questions. We would appreciate it if you could check whether your concerns are addressed properly, and update the recommendation score accordingly. We are happy to address any additional questions that you may have about our revision and rebuttal.

Authors

---

### Decision · Program_Chairs · 2023-01-20

**Decision:**

Reject

**Justification For Why Not Higher Score:**

see meta-review above

**Justification For Why Not Lower Score:**

see meta-review above

**Metareview: Summary, Strengths And Weaknesses:**

This work focuses on a novel proposed method for algorithmic recourse.

According to my evaluation and the remarks of the reviewers, one of the main weaknesses of this work is the lack of motivation, and clear exposition regarding the connection between their stated objective, and some of the theory developed. Moreover, the experimental setup was found to be somewhat weak, which made the interpretation of some of the results complicated.

**Summary Of Ac-Reviewer Meeting:**

See meta-review.

Some of the reviewers mitigated their rating (8-accept) by the inability to give a score of 7, and being discouraged to use the 5,6 scores.

There was a consensus that there was a presentation / motivation issue, and that the experiments were hard to interpret: several of us spent some time trying to dissect one of the pair of graph/table, without being able to understand it.